# Survey on Coresets for Deep Learning: Methods and Applications

## Abstract

This survey presents a comprehensive review of coreset methods in deep learning, an important tool for improving data efficiency in large-scale neural networks. In general, "coreset" is an algorithmic technique for selecting a small yet representative subset of data to replace the full dataset, which can yield more efficient training process and meanwhile preserve model performance. In the past 20 years, coreset techniques have been widely applied to many classical machine learning problems, such as clustering, regression and classification. In recent years, the coreset techniques also begin to attract a lot of attention in modern deep learning area. However, designing effective coresets usually is a challenging task since we need to take account of the trade-off among multiple different factors, such as complexity, robustness and accuracy. In this survey, we focus on two common scenarios for using coreset methods in deep learning: (1) reducing the extremely high computational cost for training a deep learning model, and (2) improving the data utilization under resource constraints such as limited label budget or storage capacity. We begin by outlining the fundamental principles, advantages, and design challenges of coresets for these two scenarios. We also discuss the emerging applications of coresets in large language models. Finally, we identify several open problems and promising directions for future research.

## 1 Introduction

In general, a *coreset* is a compact data summary designed to approximate a large dataset, enabling one to use such a data summary to complete challenging computational tasks with much lower complexities. The first formal definition of coresets dates back to Agarwal et al. (2005), where they introduce this concept in *computational geometry* for solving geometric covering and clustering problems. Over the past 20 years, the research on coresets has expanded tremendously, ranging from classical machine learning to modern deep learning. Figure 1 illustrates the technological evolution of coresets. In classical machine learning, coresets have been applied to optimization problems like clustering (Chen, 2009; Feldman & Langberg, 2011; Huang et al., 2018; Cohen-Addad et al., 2021), logistic regression (Huggins et al., 2016; Tolochinsky et al., 2022), and linear regression (Tukan et al., 2020; Huang et al., 2022a). More recently, in modern deep learning (especially in the recent 5 years), coresets have found applications in robust optimization (Ding & Wang, 2020; Huang et al., 2022b), active learning (Coleman et al., 2020; Sener & Savarese, 2018), robust training (Mirzasoleiman et al., 2020b; Dolatabadi et al., 2023), continual learning (Borsos et al., 2020; Wang et al., 2022c), and large language models (Zhang et al., 2024; Joaquin et al., 2024; Nguyen et al., 2025b).

Coresets have gained increasing importance in the deep learning era. As shown by the scaling laws (Kaplan et al., 2020), a power-law relationship exists between the improvements in model performance, intelligence and the increases in dataset size, model parameters, and computational resources (Du et al., 2022; Cherti et al., 2022). Therefore, training deep learning models often suffers from significant computational cost. For instance, it usually takes several months to train a large language model from scratch. In this context, algorithms that enable effective data compression and efficient data utilization, such as coreset-based methods, are expected to substantially reduce the computational cost (Sorscher et al., 2022; Covert et al., 2024). Specifically, we consider two major application scenarios for coresets:

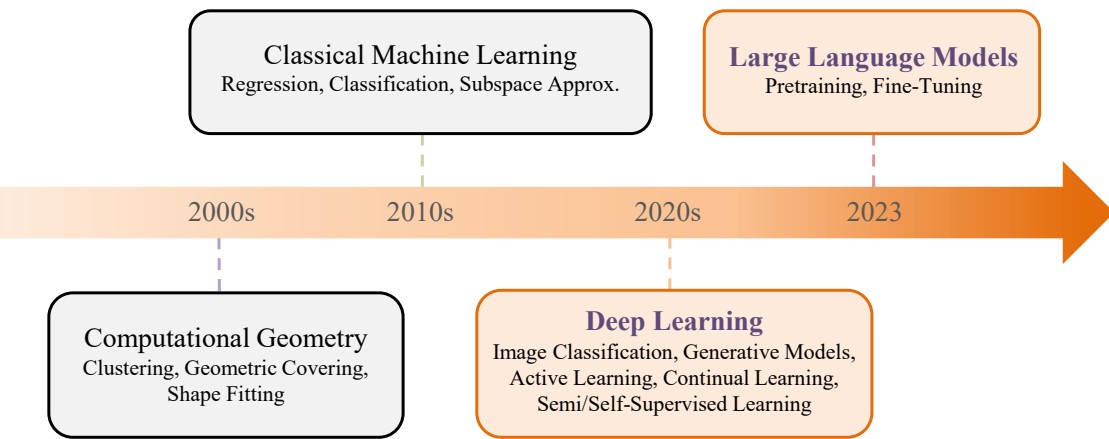

Figure 1: The technological evolution of coreset methods, progressing through four stages: coresets for computational geometry, for classical machine learning, for deep learning, and for large language models.

1. Reducing the **computational cost**, where coresets are used for compressing the original large input data. In particular, we expect that the model trained on the coreset maintains a comparable accuracy to the model trained on the original data.

2. Improving the **data utilization** under resource constraints, such as active learning (Settles, 2009) (where labeling effort and domain knowledge are limited) and continual learning (Wang et al., 2024a) (where memory and storage constraints prevent retaining all previously seen data).

To provide some intuition, we present one of the most commonly used formulations for the first scenario, i.e., using a small coreset to compress the input dataset. Let $X = \{x_i\}_{i=1}^n$ denote the original input dataset of size $n$, $Y = \{y_i\}_{i=1}^n$ denote the corresponding label vector, and $w = \{w_i\}_{i=1}^n$ denote the corresponding positive weight vector. Also, let $\Theta$ denote the parameter space of the model. Suppose $L(\theta; X, w) = \sum_{i=1}^n w_i \ell(x_i, y_i; \theta)$ is the weighted empirical risk over $X$ for the model with the parameter vector $\theta \in \Theta$ ($\ell$ is the per-sample loss). Typically, we assume that $w_i = 1$ for all $i$s, then we have the standard empirical risk, which can be simply written as $L(\theta; X) = \sum_{i=1}^n \ell(x_i, y_i; \theta)$. The objective of coreset construction is to identify a small, weighted subset $S = \{\hat{x}_i\}_{i=1}^k \subseteq X$, where the size $k \ll n$, and the corresponding positive weight vector $\hat{w} = \{\hat{w}_i\}_{i=1}^k$, serving as a compact proxy for the full dataset. Informally, we hope that the weighted loss on $S$ can approximate the loss on $X$ for all parameters in $\Theta$:

$$L(\theta; S, \hat{w}) \approx L(\theta; X), \quad \forall \theta \in \Theta. \tag{1}$$

This approximation property ensures that solving the target problem on the coreset yields a solution of quality comparable to that obtained on the full dataset. Figure 2 provides a t-SNE visualization of the coreset selection process, demonstrating how a relatively small yet representative subset can preserve the structure of the full dataset.

We should emphasize that as the rapid developments of coresets in multiple areas, their formulations become more and more diverse and it is not realistic to give a unified definition for them. The purpose of introducing (1) is simply to provide an illustrative example. The various task-specific definitions of coresets will be discussed in later sections.

Here, we should also clarify that coreset methods are related to, but distinct from, several other data reduction and selection strategies. First, coresets can be regarded as a specific approach falling under the umbrella of a broader topic called *data compression* (Lelewer & Hirschberg, 1987). In this area, another widely studied tool is *sketching*, which aims to reduce the number of features or dimensionality (Woodruff et al., 2014); the most common sketching methods include "PCA" (Maćkiewicz & Ratajczak, 1993) which uses top principal components to build low dimensional representation, and "Johnson-Lindenstrauss transforms" (Johnson et al., 1984) which projects data onto a random lower-dimensional subspace.

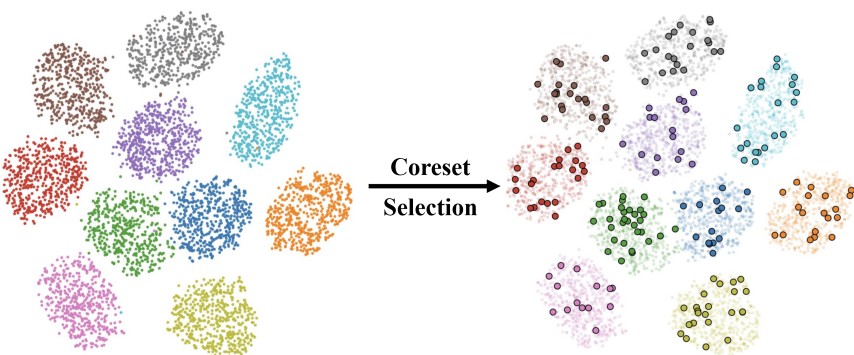

Figure 2: t-SNE (Van der Maaten & Hinton, 2008) visualization of coreset selection, which aims to construct a relatively small subset (coreset) of a large dataset. The goal is to ensure that a deep learning model trained on this coreset achieves performance comparable to training on the full dataset (as formalized in (1)). The left panel presents a t-SNE visualization of the CIFAR-10 dataset (Krizhevsky & Hinton, 2009) (all points within the same class share the same color), while the right panel shows the constructed coreset (circled points) by the method introduced in Huang et al. (2021a).

Besides sketching, coreset methods also share certain similarity with two recently proposed data reduction techniques. The technique of *dataset distillation* (Wang et al., 2018; Lei & Tao, 2023) also serves as a proxy or summarization for the original dataset in many deep learning problems. The major difference between coresets and dataset distillation is that, the former one often relies on selecting a representative subset of actual samples from the original data, but the later technique typically synthesizes a small dataset from the original training dataset which can be potentially large. The second similar technique is *data curation* (Evans et al., 2024), which aims at selecting high-quality samples from the input dataset. This process typically involves evaluating each data item using a reference model trained on a reference dataset, which specifies the target distribution to be curated. However, we need to remind the reader that **there is no strict boundary between coresets and these two similar techniques in reality.** In some cases, a data distillation or curation technique can also be regarded as a generalized coreset method in the stage of data processing.

In this article, we provide a comprehensive review of recent developments in coreset techniques and their applications in deep learning. We are aware of several excellent surveys on coreset methods, but they primarily focus on classical optimization problems (Feldman, 2020; Schwiegelshohn, 2023; Phillips, 2017; Munteanu & Schwiegelshohn, 2018), including clustering, geometric optimization (such as the smallest enclosing ball), linear and logistic regression, and subspace approximation. To the best of our knowledge, the only survey dedicated to coresets in deep learning is by Guo et al. (2022), which mainly focuses on the applications of coresets for image classification tasks.

This survey is organized as follows. Section 2 introduces four categories of commonly used coreset construction techniques for efficient model training. Section 3 explores the applications of coresets in deep learning for enhanced data utilization, with Section 4 specifically addressing their use in pretraining and fine-tuning large language models. Finally, Section 5 discusses several key challenges and promising research directions to advance the study and application of coresets in deep learning.

## 2 Coresets for Efficient Training

As we know, the *Stochastic Gradient Descent (SGD)* algorithm and a number of its variants (Qian, 1999; Duchi et al., 2011; Tieleman & Hinton, 2012; Kingma & Ba, 2015) are widely employed for training deep learning models. Despite their effectiveness, training large-scale models still entails substantial computational cost. In this section, we overview existing coreset methods for improving the efficiency of training deep learning models. Based on their underlying construction principles, we group these methods into four categories: **greedy selection, importance sampling, filtering,** and **distribution matching**. Each category employs a distinct strategy to identify the most informative training examples as the coreset. It

is worth noting that these methods are not mutually exclusive. In practice, they are often combined as a hybrid coreset approach to handle complicated tasks.

## 2.1 Greedy Selection Methods

A *greedy selection* method iteratively builds the coreset by adding data points that maximize a predefined **utility function** at each step. The core of this principle lies in designing an effective utility function that quantifies the marginal benefit of adding a new data item, which is often based on criteria like information gain, diversity, or model improvement (Hao et al., 2022; Mirzasoleiman et al., 2020a; Killamsetty et al., 2021a; Pooladzandi et al., 2022; Killamsetty et al., 2021c;b; Borsos et al., 2020; Hao et al., 2023; Zhou et al., 2022). Below, we introduce the details for this type of coreset construction methods.

Recall that $X = \{x_i\}_{i=1}^n$ represents the original dataset. For any subset of $X$, let $f : 2^X \to \mathbb{R}$ be a utility function that quantifies some measure of interest (e.g., coverage, or informativeness). A greedy selection based coreset of cardinality $k \in \mathbb{Z}^+$ is constructed via a generic iterative procedure, which is described in Algorithm 1. The algorithm is initialized with a set $S_0$, and at each iteration, it appends to the current set the element that yields the largest *marginal gain* with respect to the utility $f$. The marginal gain of adding $x \in X \setminus S$ to the current subset $S$ is defined as:

$$\Delta_f(x \mid S) = f(S \cup x) - f(S). \tag{2}$$

The procedure terminates until $|S| = k$. The resulting set $S$ is then taken as the final coreset. In deep learning, the utility function $f$ used for coreset selection is often deliberately designed to be submodular (Mirzasoleiman et al., 2020a), or to closely approximate a submodular function (Xu et al., 2023), so that the greedy selection rule in (2) can be applied efficiently with theoretical support.

A set function $f$, defined over a finite ground set $X$, is said to be *submodular* if it satisfies the following property for all subsets $A \subseteq B \subseteq X$ and any element $x \notin B$:

$$f(A \cup x) - f(A) \geq f(B \cup x) - f(B). \tag{3}$$

Inequality (3) expresses the well-known *diminishing returns property* (Fujishige, 2005), which indicates that when an element $x$ is added to a larger set $B$, the resulting gain is smaller than when it is added to a smaller set $A$. Moreover, when $f$ is *monotone* (i.e., $f(A) \leq f(B)$ for all $A \subseteq B$) and *normalized* (i.e., $f(\emptyset) = 0$), the seminal result of Nemhauser et al. (1978) (as shown in Theorem 2.1) ensures that the greedy algorithm achieves a provable approximation guarantee.

**Theorem 2.1** (Nemhauser et al., 1978). *Let $f : 2^X \to \mathbb{R}^+$ be a non-negative monotone submodular function, $k$ be the target set size, and $\{S_i\}_{i \geq 0}$ denote the sequence of subsets constructed by the greedy algorithm (Algorithm 1). Then, for all non-negative integers $l \in \mathbb{Z}^+$, the following guarantee holds:*

$$f(S_l) \geq \left(1 - \exp\left(-\frac{l}{k}\right)\right) \max_{S:|S| \leq k} f(S), \tag{4}$$

*and in particular, when $l = k$, we have*

$$f(S_k) \geq \left(1 - \frac{1}{e}\right) \max_{S:|S| \leq k} f(S). \tag{5}$$

**Remark 1.** Following Theorem 2.1, submodular maximization has been extensively studied from both complexity-theoretic and algorithmic perspectives. Exact maximization of a submodular function is NP-hard, and the $(1 - 1/e)$ approximation ratio achieved by the greedy algorithm (Algorithm 1) is provably optimal unless P = NP (Feige, 1998). Subsequent research has focused on improving computational efficiency through approximation algorithms such as *lazy greedy* (Minoux, 2005) and *stochastic greedy* (Mirzasoleiman et al., 2015). These methods often yield substantial speedups, typically one to two orders of magnitude for lazy greedy and up to three orders of magnitude for stochastic greedy, while retaining the same or nearly the same approximation guarantees.

---

**Algorithm 1** Generic greedy coreset construction

---
**Input:** Dataset $X$, utility function $f$, target size $k$
1: $S_0 \leftarrow \emptyset$
2: **for** $i = 0, \ldots, k-1$ **do**
3: $\quad x^* = \text{argmax}_{x \in X \setminus S_i} \Delta_f(x \mid S_i)$
4: $\quad S_{i+1} \leftarrow S_i \cup x^*$
5: **end for**
6: **return** $S_k$

---

A representative submodularity-based coreset construction method is **gradient matching**, which aims to align the gradient computed on the coreset with that of the full dataset (Mirzasoleiman et al., 2020a; Yang et al., 2023b; Nguyen et al., 2025a). The rationale behind this idea is that, if the gradient of the loss function over the selected subset can approximate that of the full dataset, then updating the model using the coreset should yield similar optimization dynamics. For example, Mirzasoleiman et al. (2020a) propose CRAIG (Coresets for Accelerating Incremental Gradient Descent), an algorithm designed to select a weighted subset of training data for accelerating the training of large-scale deep learning models. Specifically, the goal in CRAIG is to find the smallest subset $S \subseteq X$ and corresponding per-element stepsizes $\gamma_j > 0$ that approximate the full gradient with an error at most $\epsilon > 0$ for all possible values of the optimization parameters:

$$\min_{S \subseteq X, \, \gamma_j \geq 0 \, \forall j} |S| \quad \text{s.t.} \quad \max_{\theta \in \Theta} \left\| \sum_{x_i \in X} g_i(\theta) - \sum_{x_j \in S} \gamma_j g_j(\theta) \right\| \leq \epsilon, \tag{6}$$

where $g_i(\theta)$ denotes the gradient contribution of sample $x_i$ at parameter $\theta$, and $\Theta$ is the parameter space. They define the utility function $f(S)$ as

$$f(S) \triangleq \min_{\theta \in \Theta} \left\| \sum_{i \in X} g_i(\theta) - \sum_{j \in S} \gamma_j g_j(\theta) \right\|, \tag{7}$$

which measures the approximation error of the subset $S$ with respect to the full gradient over the entire parameter space. Then, they apply the greedy Algorithm 1 to approximately solve problem (6) by iteratively selecting the element that provides the largest marginal decrease in $f(S)$, until the approximation error is below $\epsilon$.

Besides the aforementioned submodularity-based methods, there also exist several other greedy selection approaches for coreset construction. Killamsetty et al. (2021a) propose "Grad-Match", a method that employs the orthogonal matching pursuit method (Elenberg et al., 2018) to efficiently identify coresets. Pooladzandi et al. (2022) introduce "AdaCore", a method that leverages the geometry of data distribution to extract coreset from given training examples.

Another representative greedy selection method is **loss reduction**, where the selection criterion relies on the decrease in training loss incurred when a data point is added to the coreset (Killamsetty et al., 2021c;b; Xu et al., 2023). For instance, Killamsetty et al. (2021c) introduce the method "RETRIEVE", which is a coreset selection framework designed for efficient and robust semi-supervised learning. The approach formulates the coreset selection as a mixed discrete continuous bilevel optimization problem, where the objective is to select a labeled subset that minimizes the supervised loss. Xu et al. (2023) propose a robustness-aware coreset selection (RCS) method to accelerate adversarial contrastive learning. Without relying on label information, RCS selects an informative subset by minimizing the representational divergence between natural data and their virtual adversarial variants. As exhaustively searching all possible subsets is computationally infeasible, the authors reformulate RCS as a surrogate submodular maximization problem. This transformation enables the use of a greedy algorithm, which offers an efficient solution with theoretical optimality guarantee.

## 2.2 Importance Sampling Methods

*Importance sampling* (Robert & Casella, 2004) is a popular and long-history algorithmic technique in computer science. For example, it is often used for the low-rank approximation problem (Mahoney & Drineas,

2009; Drineas et al., 2008; Mahoney et al., 2011), where the goal is to approximate a given matrix by a matrix with a lower rank. Such an approximation can be explicitly represented by a small subset of the columns and/or rows of the original matrix. These columns or rows can be found using an importance sampling based approach by first calculating an "influence score" for each one, and then sampling them with probabilities proportional to the scores. The importance sampling idea has also been introduced as an effective approach for coreset construction (Feldman & Langberg, 2011). First, it assigns an **importance score** to each data point; then, one can sample the data items according to the probability distribution derived from these scores. In contrast to uniform sampling, which treats all examples equally, this approach prioritizes data points deemed more valuable for the learning task.

Similar to the greedy selection methods introduced in Section 2.1 which rely on a predefined utility function, the key idea of importance sampling is to design a proper **scoring function** $f$ that assigns an importance score to each data point in the full dataset $X = \{x_i\}_{i=1}^n$. This score defines a proposal distribution $p$ from which samples are drawn, with the sampling probability for each $x_i \in X$ given by

$$p(x_i) = \frac{f(x_i)}{\sum_{j=1}^n f(x_j)}. \tag{8}$$

A coreset $S = \{\hat{x}_i\}_{i=1}^k$ of size $k$ is then obtained by sampling from $X$ according to $p$. To correct for the bias introduced by this non-uniform sampling, each sampled point $\hat{x}_i \in S$ is assigned a weight by

$$\hat{w}_i = \frac{q(\hat{x}_i)}{p(\hat{x}_i)}, \tag{9}$$

where $q$ is the probability density in the original dataset $X$. When $X$ is uniformly distributed, $q(x) = \frac{1}{n}$ and the weight is simply set to $\hat{w}_i = \frac{1}{n\,p(\hat{x}_i)}$. Actually, uniform sampling can be regarded as a special case where $f(x_i)$ is constant for all $i$. That is, $p(x_i) = \frac{1}{n}$.

The theoretical foundation of this importance sampling based coreset framework was established by Feldman & Langberg (2011). This framework has since been extended to numerous classical machine learning problems, including $k$-means clustering (Braverman et al., 2021), logistic regression (Munteanu et al., 2018; Tukan et al., 2020), and SVM (Tukan et al., 2020). Let $\epsilon \in (0, 1)$ be a prespecified parameter, and we use "dim" to denote the measure to evaluate the complexity of the optimization objective. For example, for a clustering problem, the dim can be defined as its *combinatorial complexity* (Feldman & Langberg, 2011). For a deep neural network, the dim can be defined as its *pseudo-dimension* (Yang et al., 2023a; Anthony & Bartlett, 2002; Bartlett et al., 2019). The value of dim is often deeply related with the *VC dimension* in learning theory (Haussler & Welzl, 1986; Vapnik & Chervonenkis, 1971; Blumer et al., 1989; Feldman & Langberg, 2011; Yang et al., 2023a; Anthony & Bartlett, 2002; Bartlett et al., 2019). According to the theory introduced in Feldman & Langberg (2011), when we construct a weighted coreset $S$ with the size $|S| = \texttt{Poly}(\frac{1}{\epsilon}, \texttt{dim})$ via the importance sampling, we can guarantee that

$$L(\theta; S, \hat{w}) \in (1 \pm \epsilon) L(\theta; X), \quad \forall \theta \in \Theta \tag{10}$$

with high probability. For more details, we refer the reader to the surveys (Feldman, 2020; Schwiegelshohn, 2023; Phillips, 2017; Munteanu & Schwiegelshohn, 2018).

In this article, we mainly focus on the applications of coresets for deep learning models. However, the theory for the aforementioned classical machine learning problems cannot be strictly guaranteed in the context of deep learning, due to its highly complicated optimization objective and non-convex loss landscapes. To shed some light, we can consider a deep neural network with piecewise linear activation functions, such as the ReLU (Nair & Hinton, 2010). It has been proven that its VC dimension has an upper bound of $O(rl \log r)$ and a lower bound of $\Omega(rl \log \frac{r}{l})$ (Bartlett et al., 2019), where $r$ and $l$ denote the numbers of parameters and layers, respectively. Therefore, the value of dim of a deep neural network is typically very large. Nevertheless, the idea of importance sampling has still been shown to be effective for improving the training efficiency of deep learning models in practice (Alain et al., 2015; Yi et al., 2019; Chang et al., 2017; Qin et al., 2024; Zheng et al., 2023b). Below, we introduce several examples.

Several methods compute the importance of each data item based on gradients. The method "Importance Sampling Stochastic Gradient Descent (ISSGD)" (Alain et al., 2015) speeds up the training procedure by reducing gradient variance in SGD. Specifically, ISSGD employs a distributed implementation that utilizes multiple worker machines to compute the gradient norms for disjoint data subsets in parallel, and then takes a master machine to collect these norms; finally, it performs importance sampling based on these gradient norms. Compared with regular SGD, which samples each batch uniformly, ISSGD can minimize training loss more quickly. For example, in the experiments on the SVHN dataset (Netzer et al., 2011), ISSGD achieves convergence in less than 1 hour, compared to about 6 hours by regular SGD (Alain et al., 2015). Instead of resorting to distributed implementations, Katharopoulos & Fleuret (2018) propose a more efficient importance sampling method through estimating upper bounds for the gradient norms. The experiments show that this approach achieves not only faster convergence but also superior model performance compared to uniform sampling.

The importance sampling idea has also been successfully adapted to accelerate the training of deep generative models, such as the *Generative Adversarial Networks (GANs)* (Goodfellow et al., 2014). The method "GAN with Flow-based Importance Sampling (FIS-GAN)" (Yi et al., 2019) performs importance sampling by prioritizing important regions in the latent noise distribution of GANs. The latent noise distribution is dynamically updated based on the Jacobian norms of the noise samples fed into the generator. The regions of noise samples with higher Jacobian norms are identified as "harder" regions, and then are assigned higher densities in the updated distribution. When training on the Fashion-MNIST dataset (Xiao et al., 2017), FIS-GAN achieves a Fréchet Inception Distance (FID) score (Heusel et al., 2017) below 3.00 in just 2000 training steps, while training without importance sampling needs 5000 training steps to reach comparable performance.

In addition to the gradient-based metrics, some methods adopt alternative criterion for measuring importance, such as prediction uncertainty (Chang et al., 2017) or training loss values (Qin et al., 2024). Chang et al. (2017) propose to prioritize training examples exhibiting high classification uncertainty, which is quantified as the variance of the predicted probability for the correct class across different training iterations. Intuitively, if the prediction for an example fluctuates significantly across iterations (high variance), it likely lies near the model's decision boundary, thereby exhibiting greater uncertainty; such uncertain examples are more important to refine the decision boundaries. InfoBatch (Qin et al., 2024) proposes measuring importance using training loss values, which can be obtained without incurring additional computational cost. Training examples with high loss values are typically not well-learned, making them informative for further training. In each training epoch, high-loss examples are retained, whereas low-loss ones are discarded according to a predefined pruning probability. In practice, InfoBatch delivers substantial training acceleration while maintaining competitive model performance. For instance, when training ResNet-50 (He et al., 2016) on the ImageNet (Deng et al., 2009) using 8 NVIDIA A100 GPUs, InfoBatch achieves 76.5% average accuracy in just 84.0 GPU hours, which is nearly 40% faster than full-data training that achieves 76.4% average accuracy in 140.0 GPU hours.

Another line of research is applying importance sampling to neural network pruning (Tukan et al., 2022; Mussay et al., 2020; 2021; Baykal et al., 2019), with the aim of compressing models rather than reducing datasets. In this setting, network parameters are treated as a large set of points, and the objective is to find a small subset that approximates the output of the original model. Unlike the methods reviewed in this section, which focus on efficient training, the coreset-based pruning methods target faster inference and broader deployment. Nevertheless, they also contribute to making deep learning more scalable and accessible for real-world applications.

## 2.3 Filtering Methods

*Filtering* methods construct coresets by discarding data points with lower importance. Unlike importance sampling introduced in Section 2.2, which introduces stochasticity through probability sampling by (8), the filtering methods are **deterministic** with hard selection rules. Typically, they set thresholds or compute rankings, to retain only those examples that meet specified standards of informativeness or relevance (Toneva et al., 2019; Paul et al., 2021; Tan et al., 2023; Sinha et al., 2020b; Xia et al., 2022; DeVries et al., 2020; Xie et al., 2025).

Formally, the coreset $S$ is defined as

$$S = \{x \in X \mid f(x) \geq \tau\}, \tag{11}$$

where $f$ is a scoring function (which could measure the uncertainty, loss, or influence) similar to the one employed by importance sampling methods, and $\tau$ is a predefined threshold. Alternatively, in the top-$k$ selection variant, the coreset consists of the $k$ highest-scoring examples.

Similar to the importance sampling methods, the choice of filtering criterion is also highly task-dependent. Toneva et al. (2019) introduce a concept called "forgetting event", which occurs when a training example, correctly classified in the previous training step, is misclassified in the current step. In their experiments on CIFAR-10, up to 35% of the dataset who have fewer forgetting events can be removed and there is only a less than 0.2% decrease in the final test accuracy. Paul et al. (2021) introduce the "Gradient Normed (GraNd)" score to quantify the importance of each individual data point. Theoretically, they show that at any given training step, the contribution of a training example to reducing the loss on any other example is bounded by its loss gradient norm (up to a multiplicative constant). This insight motivates the authors to use the GraNd score as the filtering metric, which is defined as the expected norm of the loss gradient. "Moving-one-Sample-out (MoSo)" (Tan et al., 2023) measures the importance of a training example by estimating how its removal affects the empirical risk over the remaining dataset. Intuitively, removing a critical example from the training set would significantly increase the empirical risk, while an unimportant one would have negligible impact. Directly calculating this effect is computationally prohibitive, as it requires repeatedly retraining the model with each example removed. To address this, the authors try to approximate the change in empirical risk by computing the expected inner product between the loss gradient of a candidate example and the loss gradient of the remaining dataset. An example receives a high MoSo score if its gradient strongly aligns with the gradient of the remaining dataset, indicating that training on it produces a similar effect to training on the remaining dataset.

"Top-$k$ GAN" (Sinha et al., 2020b) employs a filtering strategy for GAN training by leveraging outputs from the discriminator as importance scores. In the experiments, when training on generated images deemed least realistic by the discriminator, the resulting gradient updates tend to push them away from their nearest mode, ultimately degrading model performance and reducing training efficiency. Based on this observation, when updating the generator, Top-$k$ GAN discards such unrealistic samples by zeroing out the gradients from them. Compared to regular GAN training, Top-$k$ GAN mitigates mode dropping and enhances model performance without introducing additional computational overhead, thereby successfully improving the training efficiency.

Beyond those gradient-based and error-based importance metrics, some methods adopt different criterion, such as distance-based (Xia et al., 2022) or density-based (DeVries et al., 2020) measures. Xia et al. (2022) address the challenge of designing a scenario-agnostic scoring criteria that is robust across different task scenarios. They propose "Moderate-DS", which selects samples with scores closest to the median score, since the median is a proxy for the score distribution in statistics and could be more robust across different scenarios. Specifically, Moderate-DS first computes class centers by averaging the representations of all samples within each class, and then calculates the distance between each point and its corresponding class center. After sorting these distances, Moderate-DS constructs its coreset by selecting the points with distances closest to the median distance.

DeVries et al. (2020) accelerate GAN training by selecting data from high-density regions in the data manifold. To identify these regions, all images are first projected into an embedding space of semantical representation. A scoring function (such as the log likelihood under a standard Gaussian, or the distance to the $k$-th nearest neighbor) is then applied to calculate the density in the neighborhood of each embedded data point. Finally, only the samples with high density scores are retained for training. This sampling strategy yields remarkable training acceleration as demonstrated in their experiments. For instance, when training BigGAN (Brock et al., 2019) on the ImageNet dataset using 8 NVIDIA V100 GPUs with 16GB of RAM, the method can achieve about 4× acceleration while maintaining comparable generation performance.

Filtering methods offer a direct and easily interpretable mechanism for coreset selection. However, their deterministic nature could be a limitation since a fixed threshold has the risk of excluding underrepresented

regions in the data space. A possible solution to remedy this issue could be developing adaptive filtering strategies to balance selectivity with diversity (Wang et al., 2023b).

## 2.4 Distribution Matching Methods

The idea of *distribution matching* methods is to construct a coreset whose statistical properties are close to those of the full dataset. By minimizing the discrepancy between these two distributions, this method ensures that the selected subset retains the essential characteristics of the original data (Chakraborty et al., 2023; Zheng et al., 2023a; Ai et al., 2021; Sinha et al., 2020a; Xu et al., 2022), thereby supporting both model training and generalization.

A central challenge of distribution matching lies in selecting an appropriate distributional distance metric $d(p_X, p_S)$, where $p_X$ and $p_S$ denote the empirical distributions of the full dataset $X$ and the coreset $S$, respectively. The coreset is then obtained by solving:

$$S = \operatorname*{argmin}_{S \subseteq X, \ |S|=k} d(p_X, p_S). \tag{12}$$

A common distance metric is the Kullback-Leibler (KL) divergence:

$$d_{\mathrm{KL}}(p_S \| p_X) = \int p_S(x) \log \frac{p_S(x)}{p_X(x)} \ \mathrm{d}x. \tag{13}$$

"Bayesian Pseudo-coresets Exemplar VAE (ByPE-VAE)" (Ai et al., 2021) employs a distribution matching method based on the KL-divergence (13) to speed up the training of Exemplar VAE (Norouzi et al., 2020). Exemplar VAE is a variant of *Variational Autoencoders (VAEs)* (Kingma & Welling, 2014) with a mixture of exemplar-based priors on the latent space. Specifically, a set of exemplars is chosen from the training set, and an exemplar-based prior is learned for each exemplar. The size of the exemplar set could be potentially as large as the training set, which makes Exemplar VAE computationally expensive to train. To address this limitation, ByPE-VAE employs Bayesian pseudo-coresets (Manousakas et al., 2020) to create a small-scale pseudo-coreset of pseudo-data points, which is constructed by minimizing the KL-divergence (13) between the posterior distributions induced by the pseudo-coresets and the true data points. By using these pseudo-data points as exemplars, ByPE-VAE achieves significant training accelerations without sacrificing model performance. For instance, on Fashion-MNIST and CIFAR-10 using a single NVIDIA 1080Ti GPU, ByPE-VAE delivers approximately $3\times$ faster training compared to standard Exemplar VAE while maintaining comparable performance.

Besides the approaches that try to directly minimize a distributional distance metric, there also exist several "indirect" distribution matching methods. They usually employ other techniques to implicitly minimize the discrepancy between the coreset and the input distribution. One common indirect approach is clustering-based methods. These methods use clustering techniques, such as $k$-center clustering (Gonzalez, 1985), to try to minimize the discrepancy by selecting the most representative data items (Sinha et al., 2020a; Xu et al., 2022).

In each iteration of GAN training, "Small-GAN" (Sinha et al., 2020a) tries to increase the effective batch size by formulating it as a $k$-center problem, with the Euclidean distance as the distance metric. Coresets are constructed by first randomly sampling a larger batch, and then using the Gonzalez's algorithm (Gonzalez, 1985) to select a smaller batch from the larger one. "Fréchet Descriptor Distance based Coreset (FDD-Coreset)" (Xu et al., 2022) also formulates coreset selection as a $k$-center problem, but employs the Fréchet distance (Dowson & Landau, 1982) as the distance metric.

## 2.5 Summary

We have introduced four categories of coreset construction methods for efficient model training. Submodularity-based greedy selection methods enjoy strong theoretical guarantees. They can produce quality guaranteed coresets that provide a $(1 - 1/e)$-approximation to the optimal solution, as shown in Theorem 2.1. Filtering methods strictly discard low-score examples and focus exclusively on challenging or

informative ones. As a result, coresets constructed by them tend to lack diversity. In contrast, importance sampling methods naturally introduce diversity since they select probabilistically rather than deterministically. However, they may select less informative examples from unimportant regions in the data distribution. Another advantage of importance sampling is that it can provide an unbiased estimator of the loss on the full dataset through reweighting by (9). Distribution matching methods can select representative examples since they aim to preserve the overall data distribution. However, similar to importance sampling methods, they may select examples that are easy or already well learned.

In addition to the coresets for efficient training introduced in this section, these four categories of methods are also applied to other scenarios, including coresets for enhancing data utilization introduced in Section 3 and for large language models in Section 4. For example, the "SAS" (Joshi & Mirzasoleiman, 2023) and the "SimCore" (Kim et al., 2023a) methods for semi/self-supervised learning introduced in Section 3.3.3, and the "CoLM" method (Nguyen et al., 2025b) for large language models in Section 4.2, all formulate coreset selection as a submodular optimization problem. Xie et al. (2023) use the idea of importance sampling to select data for pretraining large language models, as introduced in Section 4.1. While many active learning algorithms introduced in Section 3.1 act as filtering methods since they assign scores to examples and select those with the highest scores (Yoo & Kweon, 2019; Mayer & Timofte, 2020; Ducoffe & Precioso, 2018; Geng et al., 2023; Wang et al., 2022b; Kim et al., 2023b), some algorithms adopt alternative strategies such as importance sampling or distribution matching. For instance, the "DACS" method (Kim & Shin, 2022) introduced in Section 3.1.3 selects more samples from the sparse regions in the data distribution and less from the dense regions, which is similar to the idea of importance sampling. And the "WAAL" method (Shui et al., 2020) in Section 3.1.3 explicitly formulates active learning as a distribution matching problem. Please see the corresponding sections for more details. Overall, these cases demonstrate that the four categories of methods introduced in this section have broad applicability across different coreset selection scenarios.

## 3 Coresets for Enhancing Data Utilization with Limited Resource

In this section, we survey the coreset selection techniques developed for enhancing data efficiency. In particular, we review the methods for three scenarios with limited resource: **active learning**, **continual learning** and **semi/self-supervised learning**. In active learning, coreset selection is used to identify the most informative unlabeled data points for annotation, so as to avoid spending too much human labeling effort (Settles, 2009). In continual learning, a central challenge is the "catastrophic forgetting", where the model could lose knowledge from previously learned tasks when trained on new ones (McClelland et al., 1995; McCloskey & Cohen, 1989). A common approach for addressing this problem is to store a small subset (i.e., the coreset) of past data in a replay buffer (Buzzega et al., 2020; Aljundi et al., 2019). Self-supervised learning (Gui et al., 2024) and semi-supervised learning (Yang et al., 2022) consider the scenario that we are given a large pool of unlabeled data, and there is no labeled data or the amount of labeled data is quite limited. For semi/self-supervised learning, a key difference with the aforementioned active learning is that we are not allowed to draw support from human effort to add labels.

### 3.1 Active Learning

In this section, we focus on **pool-based active learning**, which is the most widely adopted type of active learning in deep learning. Pool-based active learning assumes that there is an initial small labeled dataset and a large pool of unlabeled data available for sampling. It iteratively performs the following four steps: (1) training the target model on the labeled dataset, (2) selecting important unlabeled examples based on the trained model, (3) having the selected examples labeled by annotators, and (4) adding the newly labeled data into the labeled dataset, as shown in Figure 3. Typically, a labeling budget is imposed, which is much smaller than the size of the unlabeled dataset. In this procedure, only a small number of important examples are labeled within the budget, and it is natural to consider it from the perspective of coreset construction. For instance, Sener & Savarese (2018) have demonstrated that the problem of active learning for convolutional neural networks can be formulated as a coreset selection problem.

Next, we treat the data selection strategies of active learning algorithms as coreset selection methods and provide a detailed review of them. Based on the criteria for evaluating the importance of unlabeled exam-

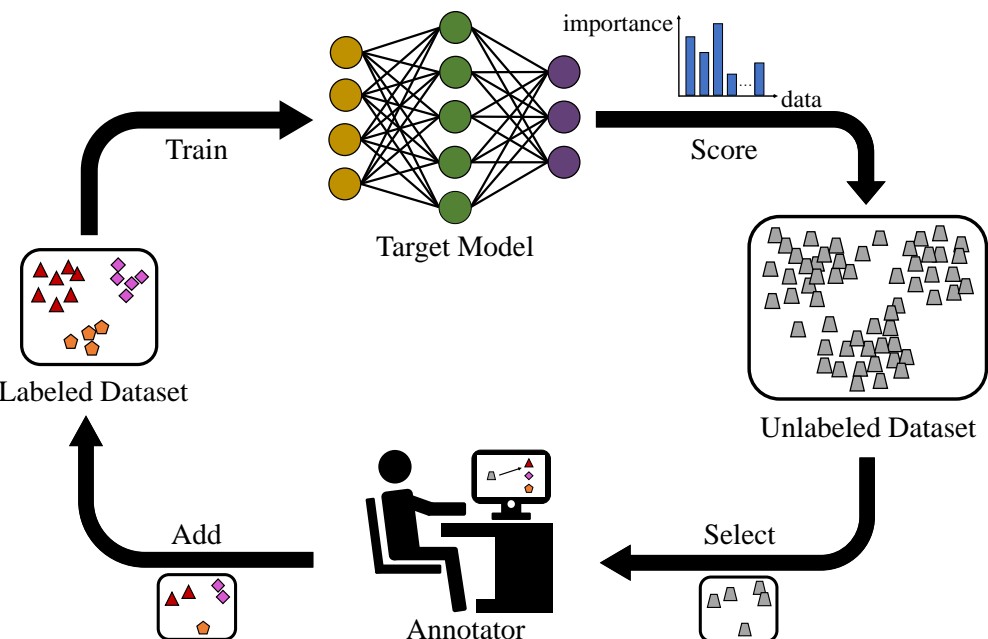

Figure 3: Pool-based active learning repeatedly executes four key steps: (1) training the target model using the labeled data, (2) scoring the importance of unlabeled examples and selecting the most important ones, (3) obtaining annotations for the selected examples from annotators, and (4) incorporating the newly labeled data into the existing labeled dataset.

ples, we categorize active learning approaches into three types: **loss-based**, **coverage-based**, and **hybrid methods**. A summary of several representative methods from each category is presented in Table 1.

Note that our goal is not to provide an exhaustive survey of active learning, but to shed light on the key strategies for assessing and identifying important unlabeled examples from the perspective of coresets construction. For more comprehensive introductions on active learning, we refer the reader to the surveys (Settles, 2009; Ren et al., 2021; Liu et al., 2022; Li et al., 2024a).

### 3.1.1 Loss-Based Methods

Loss-based methods prioritize unlabeled examples that are expected to yield high training loss if added to the training set. Such examples typically include those with high predicted training loss (Yoo & Kweon, 2019; Huang et al., 2021b) or high prediction uncertainty (Wang et al., 2016; Ranganathan et al., 2017; Ducoffe & Precioso, 2018; He et al., 2019; Mayer & Timofte, 2020; Kim et al., 2021b; Jung et al., 2023; Kye et al., 2023; An et al., 2024; Gao et al., 2020; Parvaneh et al., 2022; Geng et al., 2023). By concentrating on the examples where the model performs poorly or exhibits high uncertainty, these methods try to improve the model by training it on the most challenging or informative examples. Instead of focusing on training loss, some other loss-based approaches consider generalization performance, prioritizing unlabeled examples that are predicted to maximally reduce the test-time error (Ash et al., 2021; Wang et al., 2022b;a; Kim et al., 2023b). Next, we introduce these methods in more detail.

To identify unlabeled examples that are likely to produce high **training loss**, Yoo & Kweon (2019) design a loss prediction model to estimate the loss for unlabeled examples and select those with the highest predicted losses for annotation. Instead of relying on an auxiliary loss prediction model, Huang et al. (2021b) demonstrate that the training loss for an unlabeled example can be estimated via the difference between two consecutive active learning iterations.

Rather than directly estimating training loss, many methods select unlabeled examples for which the model exhibits the highest **uncertainty**. These uncertain examples usually result in high training loss because the

Table 1: **Representative coreset selection methods in active learning**. The "Availability" column indicates the public accessibility of the corresponding code or implementation: "official" refers to code released by the original authors, "third-party" refers to unofficial implementations by independent developers, and "N.A." means no publicly available code has been found (to the best of our knowledge).

| Type | Methods | Highlights | Availability |
|---|---|---|---|
| Loss-Based | CEAL (Wang et al., 2016) | Selecting examples using least confidence (14), margin (15), or entropy (16), querying true labels for some and assigning pseudo-labels to others. | third-party (1, 2, 3) |
| | DFAL (Ducoffe & Precioso, 2018) | Selecting examples closest to decision boundaries. | official |
| | Yoo & Kweon (2019) | Training a model to predict loss for unlabeled examples. | third-party (1, 2, 3) |
| | Gao et al. (2020) | Prioritizing examples with inconsistent predictions under data augmentations. | third-party (1) |
| | SAAL (Kim et al., 2023b) | Selecting examples exhibiting the highest training loss sharpness. | official |
| Coverage-Based | Sener & Savarese (2018) | Formulating active learning as a $k$-center problem. | official |
| | DAL (Gissin & Shalev-Shwartz, 2019) | Employing a binary classifier to discriminate between labeled and unlabeled examples. | official |
| | VAAL (Sinha et al., 2019) | Adversarially training a discriminator to discriminate between labeled and unlabeled examples. | official |
| | ProbCover (Yehuda et al., 2022) | Formulating active learning as a Max Probability Cover problem. | official |
| | MaxHerding (Bae et al., 2024) | Selecting examples that maximizes their similarity to the entire unlabeled dataset. | N.A. |
| Hybrid | BADGE (Ash et al., 2020) | Selecting examples by applying $k$-means++ on their gradient embeddings. | official |
| | WAAL (Shui et al., 2020) | Formulating active learning as a distribution matching problem. | official |
| | NoiseStability (Li et al., 2024c) | Selecting examples based on the deviations in their feature embeddings under small perturbations. | official |
| | Cluster-Margin (Citovsky et al., 2021) | Performing hierarchical agglomerative clustering and selecting examples with low margin scores (15) from the clusters. | third-party (1, 2) |
| | LDM-S (Cho et al., 2024) | Selecting high-diversity examples closest to decision boundaries. | official |

model lacks sufficient knowledge to predict their labels confidently. In image classification tasks, common uncertainty measures include *least confidence*, *margin*, and *entropy*. Given an unlabeled dataset $X$, least confidence based methods select the example with the lowest confidence in its top predicted class:

$$x^* = \operatorname*{argmax}_{x \in X} \; 1 - P_\theta(\hat{y}|x), \tag{14}$$

where $P_\theta(y|x)$ is the conditional probability of class $y$ given input $x$, under the model parameterized by $\theta$, and $\hat{y} = \operatorname{argmax}_y P_\theta(y|x)$. One limitation of least confidence is that it only considers the class with the highest probability, ignoring the probability distribution over other classes. To alleviate this problem, margin based methods take into account the probabilities of the top two most probable classes:

$$x^* = \operatorname*{argmin}_{x \in X} \; P_\theta(\hat{y}_1|x) - P_\theta(\hat{y}_2|x), \tag{15}$$

where $\hat{y}_1 = \operatorname{argmax}_y P_\theta(y|x)$ and $\hat{y}_2 = \operatorname{argmax}_{y \neq \hat{y}_1} P_\theta(y|x)$. Entropy is a more general measure that leverages the information from all classes:

$$x^* = \operatorname*{argmax}_{x \in X} \; -\sum_i P_\theta(y_i|x) \log P_\theta(y_i|x). \tag{16}$$

The method "Cost-Effective Active Learning (CEAL)" (Wang et al., 2016) uses those three uncertainty measures (defined in (14), (15), and (16), respectively) to select unlabeled examples for annotation, and

incorporates low-entropy unlabeled examples into the training set by assigning pseudo-labels to them. Ranganathan et al. (2017) not only employ the entropy (16) to select unlabeled examples but also integrate it into the training loss of the target model. The method "Adversarial Sampling for Active Learning (ASAL)" (Mayer & Timofte, 2020) employs the entropy (16) in a different way by optimizing a generator to produce a synthetic example that maximizes classification entropy. Then, the unlabeled example closest to the synthetic example is selected and annotated. "Look-Ahead Data Acquisition (LADA)" (Kim et al., 2021b) integrates data augmentation into active learning by learning an augmentation policy through maximizing the classification entropy (16). "Training Dynamics for Active Learning (TiDAL)" (Kye et al., 2023) computes the margin (15) and entropy (16) based on the training dynamics of unlabeled examples, which is defined as their averaged classification results across training iterations. An et al. (2024) propose to consider the reduction of classification entropy (16) for the remaining unlabeled examples when a specific unlabeled example is labeled. The entropy and its reduction are combined through a convex combination, which is then used to select the most informative unlabeled examples for annotation.

An alternative measure of uncertainty for an unlabeled example is its **distance to the decision boundary** of the model. Examples closer to the decision boundary exhibit higher uncertainty, since they lie in regions where even small perturbations could lead to a change in their predicted class. Since calculating the exact distance is intractable, the approach "DeepFool Active Learning (DFAL)" (Ducoffe & Precioso, 2018) tries to approximate the distance by determining the smallest perturbation applied to an unlabeled example that is needed to reclassify it into a different category. Gao et al. (2020) prioritize unlabeled examples with inconsistent predictions under data augmentations. "Active Learning by FeAture Mixing (ALFA-Mix)" (Parvaneh et al., 2022) selects unlabeled examples exhibiting prediction inconsistency when their feature embeddings are interpolated with classwise average features of labeled examples. The method "Multi-classifier Adversarial Optimization for Active Learning (MAOAL)" (Geng et al., 2023) employs adversarial training to simultaneously train a feature generator and multiple classifiers, and selects unlabeled examples with the highest classification discrepancy across the classifiers for annotation.

While most approaches rely on training loss for identifying important unlabeled examples, there are also some methods that select examples based on their predicted contribution to reducing the **generalization error** of the target model. "Batch Active learning via Information maTrices (BAIT)" (Ash et al., 2021) selects unlabeled examples by minimizing the expected log-likelihood error on the whole unlabeled dataset through optimizing an objective involving the Fisher information matrix (which is the Hessian of the negative log-likelihood function) (Chaudhuri et al., 2015). Wang et al. (2022b) illustrate that selecting unlabeled data with the highest gradient norm can maximally boost the testing performance of the target model. The method "dynamicAL" (Wang et al., 2022a) leverages the training dynamics of unlabeled examples, which is defined as the derivative of the training loss with respect to the training iteration, to select valuable examples. "Sharpness-Aware Minimization (SAM)" (Foret et al., 2021) demonstrates that the generalization error is upper bounded by the maximal training loss sharpness in addition to a parameter regularization term. Therefore, SAM trains models by simultaneously minimizing the loss value and loss sharpness. Inspired by this, another method "Sharpness-Aware Active Learning (SAAL)" (Kim et al., 2023b) estimates the loss sharpness for unlabeled examples by assigning pseudo-labels and selects those with the highest estimated sharpness for annotation.

### 3.1.2 Coverage-Based Methods

Coverage-based methods aim to select unlabeled examples that best represent the underlying data distribution. This is often achieved by prioritizing examples that either maximize coverage of the data space (Yehuda et al., 2022; Bae et al., 2024; 2025; Hua et al., 2025) or enhance diversity of the selected set (Geifman & El-Yaniv, 2017; Sener & Savarese, 2018; Gissin & Shalev-Shwartz, 2019; Sinha et al., 2019; Agarwal et al., 2020; Kim et al., 2021a; Kothawade et al., 2021; Hacohen et al., 2022).

**Coverage maximization methods.** These methods select unlabeled examples for covering different regions of the data space, where the goal is to capture a broad range of the overall data distribution. "Probability Coverage (ProbCover)" (Yehuda et al., 2022) formulates it as a Max Probability Cover problem. Based on this formulation, ProbCover selects unlabeled examples maximizing the probability of the union of balls centered at each selected point with a predefined radius. "MaxHerding" (Bae et al., 2024) generalizes

ProbCover by introducing a notion of generalized coverage, which is defined as the expected maximum similarity between the labeled set and the overall data distribution. Conceptually, it encourages to select the unlabeled examples that maximize their similarity to the entire unlabeled dataset. One limitation of MaxHerding is that it exclusively focuses on coverage but neglects uncertainty. To remedy this issue, the recent method "Uncertainty Herding (UHerding)" (Bae et al., 2025) extends the generalized coverage concept by integrating it with an uncertainty measure. More hybrid methods that simultaneously consider coverage and uncertainty are introduced in Section 3.1.3.

**Diversity maximization methods.** Rather than directly optimizing for coverage, many methods seek to select unlabeled examples that are distinct from each other. Intuitively, a **diverse set** of examples can reduce information redundancy, ensuring that the target model encounters a broader range of features or patterns during training. "Farthest First Active learning (FF-Active)" (Geifman & El-Yaniv, 2017) selects the next point as the one farthest from the already selected points in the feature space. Sener & Savarese (2018) formulate active learning for convolutional neural networks as a $k$-center clustering problem in feature space, and use the Gonzalez's algorithm (Gonzalez, 1985) to greedily select an initial subset of data points. Then, a mixed integer program is employed to further refine the subset. The method "Contextual Diversity based Active Learning using Core-Sets (CDAL-CS)" (Agarwal et al., 2020) also uses the Gonzalez's algorithm, but employs a notion of "pairwise contextual diversity" as the distance metric (rather than the distance induced in feature space).

Instead of relying on explicit distance metrics to select diverse examples, several active learning methods employ auxiliary classifiers (Gissin & Shalev-Shwartz, 2019) or adversarial training (Sinha et al., 2019; Kim et al., 2021a) to implicitly quantify their dissimilarities. "Discriminative Active Learning (DAL)" (Gissin & Shalev-Shwartz, 2019) trains a binary classifier to differentiate between unlabeled and labeled examples. The unlabeled examples classified as unlabeled with the highest confidence are then selected. Another approach "Variational Adversarial Active Learning (VAAL)" (Sinha et al., 2019) identifies important unlabeled examples through an adversarial game between a VAE and a discriminator. The VAE is trained to deceive the discriminator into classifying both labeled and unlabeled examples as labeled, while the discriminator is trained to differentiate between them. After this training procedure, unlabeled examples classified as unlabeled with the highest confidence by the discriminator are selected. One limitation of VAAL is its task-agnostic nature, which overlooks task-specific information. To address this limitation, an improved approach "Task-Aware Variational Adversarial Active Learning (TA-VAAL)" (Kim et al., 2021a) integrates VAAL with a task-aware method which trains a ranking model to prioritize examples by their loss rankings.

Recall that in Section 2.1, we introduced the definition of a utility function $f : 2^X \to \mathbb{R}$ defined over a given dataset $X$, with the property of submodularity. A submodular utility function $f$ can be used for measuring similarity due to the diminishing returns property (3). This property can be interpreted from the perspective of *information gain* by viewing $f(S)$ as the amount of information contained in $S$. From this perspective, the property (3) states that the information gained by adding a new element to a set decreases as the set grows. This is because a larger set is more likely to already contain information that is similar to that of the new element. Therefore, maximizing a proper submodular utility function naturally encourages the selection of diverse and representative examples and discourages redundant ones. "Submodular Information Measures based actIve LeARning (SIMILAR)" (Kothawade et al., 2021) employs submodular information measures (Iyer et al., 2021; Kothawade et al., 2022) to deal with the bias issue in real scenarios, including datasets with imbalanced classes, out-of-distribution data points, or high redundancy. For example, the *submodular mutual information (SMI)* is used to select unlabeled examples when datasets contain rare classes (Kothawade et al., 2021). Given an unlabeled dataset $X$, a set $R$ consisting of a small number of examples from the rare classes, and a submodular function $f$ that measures the diversity of an input set, a set $S \subseteq X$ can be constructed by maximizing the SMI, which is defined as:

$$I_f(S; R) = f(S) + f(R) - f(S \cup R). \tag{17}$$

Since $R$ is a fixed set, $f(R)$ is also fixed. Therefore, maximizing (17) is equivalent to selecting a set $S$ that is not only diverse itself (high $f(S)$) but also similar to $R$ (low $f(S \cup R)$). In other words, it encourages to select a diverse set that contains examples similar to the known rare ones.

### 3.1.3 Hybrid Methods

There also exist numerous "hybrid" methods that jointly optimize uncertainty and diversity, aiming at identifying examples that are both informative and representative (Bae et al., 2025; Yin et al., 2017; Shui et al., 2020; Ash et al., 2020; Wang et al., 2020; Citovsky et al., 2021; Kim & Shin, 2022; Li et al., 2024c; Cho et al., 2024). That is, the joint optimization enables an effective trade-off between exploitation (refining the model in uncertain regions) and exploration (probing regions with unseen patterns).

Yin et al. (2017) take a two-step approach: first a set of points is selected based on the classification entropy (16) and dissimilarity, then a set of additional points are selected based solely on dissimilarity. Similar with the coreset idea introduced in Section 2.4, "Wasserstein Adversarial Active Learning (WAAL)" (Shui et al., 2020) formulates the active learning problem as a distribution matching problem between the underlying data distribution and the distribution of queried data.

Clustering algorithms play an important role in active learning as they are often used to simultaneously optimize both uncertainty and diversity. The approach "Batch Active learning by Diverse Gradient Embeddings (BADGE)" (Ash et al., 2020) runs $k$-means++ (Arthur & Vassilvitskii, 2007) to select unlabeled examples, where the distance metric is based on their gradient embeddings computed from pseudo-labels and last-layer network parameters. "Cluster-Margin" (Citovsky et al., 2021) performs hierarchical agglomerative clustering on unlabeled examples and focuses on the clusters containing examples with low margin scores (15). These clusters are sorted in ascending order according to their size. Then, Cluster-Margin samples from the sorted clusters in a round-robin scheme until the labeling budget is exhausted. "Density-Aware Core-Set (DACS)" (Kim & Shin, 2022) shows that examples from lower-density regions exhibit higher prediction entropy (16) and loss, suggesting that unlabeled examples from sparse regions are more informative. Therefore, DACS clusters unlabeled examples by their estimated densities and samples from each cluster inversely proportional to its size (which is similar to the idea of importance sampling introduced in Section 2.2). "Least Disagree Metric based Sampling (LDM-S)" (Cho et al., 2024) introduces Least Disagree Metric (LDM) to quantify the distance of an example to the decision boundary. Then, it incorporates a modified version of $k$-means++ to promote diversity, ensuring that the selected examples are both sufficiently informative and representative.

## 3.2 Continual Learning

*Continual learning* (Kirkpatrick et al., 2017; Wang et al., 2024a) is a learning paradigm in which artificial neural networks are trained on a sequence of tasks, aiming to acquire new knowledge while preserving performance on previously learned tasks. This paradigm faces a fundamental challenge known as **catastrophic forgetting** (McClelland et al., 1995; McCloskey & Cohen, 1989), where adapting to new tasks leads to significant degradation in performance on earlier ones. Replay-based methods (Buzzega et al., 2020; Aljundi et al., 2019; Lin et al., 2024a) have been widely studied for mitigating catastrophic forgetting. These methods typically maintain a small buffer that contains a subset of past training samples (see Figure 4 for an illustration). Given the limited storage capacity, the key challenge lies in how to construct the **memory buffer**. It is natural to consider this problem from the perspective of coreset selection, which is to identify the most representative samples (similar with active learning in Section 3.1). Formally, replay-based methods operate under the constraint of a fixed-size memory buffer $M$. For each $t \geq 1$, let $X_t$ be the data of the $t$-th task, and the goal is to select a coreset $S_t \subseteq X_t$ that is able to approximate $X_t$. The selected coreset $S_t$ is stored in the memory buffer $M$. When training on a new task, the parameters $\theta$ of the model are updated by optimizing a composite objective function (Tiwari et al., 2022), which incorporates both the current task data and the stored coresets:

$$L(\theta) = L(\theta; X_{t+1}) + \lambda L(\theta; M), \tag{18}$$

where $\lambda > 0$ controls the rehearsal strength. In this section, we categorize the coreset selection methods in continual learning into two major types: coverage-based methods and gradient-based methods (see Table 2 for an illustration).

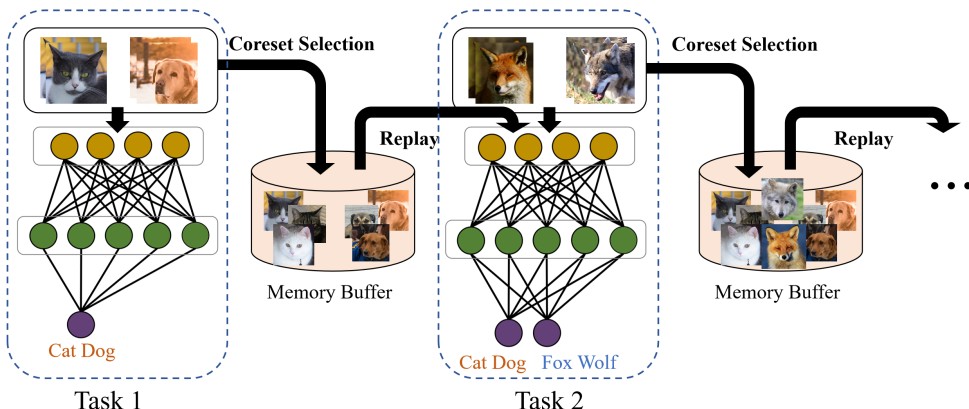

Figure 4: Overview of the replay-based methods for continual learning. This approach aims to retain the learned knowledge from previous tasks by storing a small subset of past training samples in a limited *memory buffer*.

Table 2: **Coreset selection methods in continual learning**. The "Availability" column indicates the public accessibility of the corresponding code or implementation: "official" refers to code released by the original authors, and "N.A." means no publicly available code has been found (to the best of our knowledge).

| Type | Methods | Highlights | Availability |
|---|---|---|---|
| Coverage-Based | Reservoir Sampling (Riemer et al., 2019) (Buzzega et al., 2020) (Boschini et al., 2022) | A fundamental streaming algorithm for maintaining fixed-size uniform samples from unbounded data streams of unknown cardinality. | official |
| | Ring Buffer (Lopez-Paz & Ranzato, 2017) (Chaudhry et al., 2019a) (Chaudhry et al., 2021) | A class-partitioned memory architecture designed to ensure balanced exemplar retention. | N.A. |
| | $k$-Means (Chaudhry et al., 2019b) | Using online $k$-means to estimate the $k$ class centroids in the penultimate feature space and storing the input samples closest to each centroid in memory. | official |
| Gradient-Based | GSS (Aljundi et al., 2019) | Formulating coreset selection as a constraint optimization problem of continual learning. | official |
| | OCS (Yoon et al., 2022) | A coreset selection method that selects the most representative and informative coreset at each iteration and trains them in an online manner. | official |
| | Greedy Coreset (Borsos et al., 2020) | Proposing a coreset construction via cardinality-constrained bilevel optimization. | official |
| | BCSR (Hao et al., 2023) | A new bilevel formulation where the inner problem minimizes expected training error from a sampled distribution, and the outer problem learns a sparse distribution with $K$ nonzero entries to minimize overall training error. | official |
| | GCR (Tiwari et al., 2022) | A method for selecting a coreset that approximates the model parameter gradients over all previously seen data. | N.A. |
| | RM (Bang et al., 2021) | A method that emphasizes the importance of maintaining sample diversity in coreset. | official |

### 3.2.1 Coverage-Based Methods

The methods in the first category usually maintain predefined selection mechanisms and do not adapt dynamically during training procedure (in contrast to the gradient-based methods discussed in Section 3.2.2). Representative methods include reservoir sampling (Riemer et al., 2019; Buzzega et al., 2020; Boschini et al., 2022), ring buffer (Lopez-Paz & Ranzato, 2017; Chaudhry et al., 2019a; 2021), and $k$-means clustering (Chaudhry et al., 2019b).

*Reservoir sampling* (Vitter, 1985) is a foundational streaming algorithm for maintaining fixed-size uniform samples from a data stream with unknown cardinality. The canonical implementation initializes a reservoir $M$ with the first $k$ observed samples. For the $i$-th incoming sample $(i > k)$, the algorithm retains it with probability $P_i = k/i$, replacing a uniformly random element in $M$ if selected. This probabilistic replacement rule ensures that every observed sample has an equal probability of $k/n$ to be selected, where $n$ denotes the total number of elements in the stream. The reservoir sampling algorithm achieves $O(n)$ time complexity and $O(k)$ space complexity, making it asymptotically optimal for single-pass scenarios. Due to these properties, it is particularly suitable for the applications involving large-scale data streams.

The integration of reservoir sampling into continual learning originated with Riemer et al. (2019), who pioneered its use for task-agnostic memory buffering. Their method investigates the continual learning problem from the balance between learning plasticity and memory stability, which is achieved through gradient alignment, i.e., by encouraging gradients of different tasks to point in similar directions so that learning on one task facilitates rather than interferes with others. They introduce the method "Meta Experience Replay (MER)", which integrates experience replay with meta-learning to enhance knowledge transfer while mitigating interference.

In some continual learning scenarios, we often encounter multi-label datasets that exhibit long-tailed distributions, where a few dominant classes are overrepresented while many minority classes have limited samples. To address this challenge, Kim et al. (2020) propose a sampling strategy termed "Partitioning Reservoir Sampling (PRS)", which is designed to allocate a proportional share of the memory buffer to moderate and minority classes, thereby preserving a more balanced representation for both current and past experiences. Moreover, Chrysakis & Moens (2020) introduce a strategy called "Class-Balancing Reservoir Sampling (CBRS)" to mitigate the imbalance issue.

In addition to employing reservoir sampling for storing exemplars, recent works (Buzzega et al., 2020; Boschini et al., 2022) propose to retain some intermediate products of training, e.g., the logits. Specifically, Buzzega et al. (2020) introduce "Dark Experience Replay (DER)", which leverages reservoir sampling to match the current logits with those stored from past experiences so that the model remains consistent with previously learned knowledge. However, Boschini et al. (2022) demonstrate that DER may suffer from several limitations. For instance, it can introduce classification bias and overlook important semantic relationships between previously learned and newly introduced classes. To overcome these drawbacks, they propose eXtended-DER (X-DER), a novel continual learning method that embraces memory update and future preparation. In particular, they first introduce a future preparation strategy, which exploits past and present data to prepare future classification heads so that they can better encode meaningful information. Then, they propose a memory update procedure that keeps the memory buffer up to date by inserting secondary information from the present into memories of the past.

The *ring buffer* strategy was first introduced by Lopez-Paz & Ranzato (2017) and subsequently adopted in later studies on continual learning (Chaudhry et al., 2019a; 2021). Broadly speaking, the ring buffer maintains a fixed-size memory that stores a subset of past samples and updates this set in a first-in-first-out (FIFO) manner as new data arrive. In this context, the strategy is applied to efficiently manage and refresh stored examples during training, which allows the model to rehearse previous knowledge while keeping memory usage constant. Specifically, given $C$ total classes and a memory buffer $M$, each class $c \in \{1, 2, \ldots, C\}$ is assigned a buffer $M^c$ with a quota of

$$|M^c| = \left\lfloor \frac{|M|}{C} \right\rfloor$$

under a FIFO replacement policy. Formally, for each incoming batch $B$ of data points, let $B^c = \{x \in B \mid y = c\}$ denote the set of points in $B$ belonging to class $c$, where $y$ is the label of $x$. Also, let $m^c$ denote the number of currently stored points in the buffer $M^c$. Then, each $M^c$ is updated as follows:

$$M^c = \begin{cases} \text{FIFO}\big(M^c \cup B^c\big) & \text{if } |M^c| - m^c < |B^c|, \\ M^c \cup B^c & \text{otherwise.} \end{cases}$$

This deterministic allocation guarantees equal buffer capacity across classes, ensuring a balanced representation of all classes in memory.

The *k-means* algorithm, which is adopted by Chaudhry et al. (2019b), provides a geometric framework for coreset construction in continual learning. For each class $c \in \{1, \ldots, C\}$, the method performs an online variant of the Lloyd's algorithm (a popular implementation for $k$-means (Ostrovsky et al., 2012)) to iteratively estimate $k^c = \lfloor |M|/C \rfloor$ centroids $\{\mu_i^c\}_{i=1}^{k^c}$ in the latent feature space (preceding the classification layer of the model). This strategy shares similar advantages and limitations with the ring buffer approach, but yields improved coverage of the constructed coreset in feature space.

### 3.2.2 Gradient-Based Methods

The second category of approaches for continual learning is gradient-based, which leverages the information from gradients to guide coreset selection. A pioneering work in this direction is "Gradient-based Sample Selection (GSS)" by Aljundi et al. (2019), who formulate the sample selection problem as the following optimization problem:

$$\theta_t = \operatorname*{argmin}_{\theta} L(\theta; X_t) \quad \text{s.t.} \quad L(\theta; X_i) \leq L(\theta_{t-1}; X_i); \quad \forall i \in [0 \ldots t-1], \tag{19}$$

where $\theta_{t-1}$ is the model trained on task $t-1$, $t$ denotes the index of the current task, and $i$ ranges over the previously seen tasks. The objective is to minimize the loss on the current examples without increasing the losses on the previously learned examples, which effectively constrains parameter updates to preserve performance on past tasks and thereby mitigates catastrophic forgetting. Meanwhile, the authors demonstrate that this selection process is equivalent to maximizing sample diversity in the replay buffer and propose a lightweight greedy algorithm to solve it. However, this method does not consider the imbalance and noisy issues which are common in real-world data streams.

To address these issues, Yoon et al. (2022) propose "Online Coreset Selection (OCS)", a simple yet effective method which selects the most representative and informative samples at each iteration, and trains the model on them in an online fashion. Specifically, they introduce three gradient-based selection criterion for coresets: (1) minibatch similarity, which selects the samples that are most representative of the current task distribution; (2) sample diversity, which encourages selecting the samples with minimal redundancy to improve coverage of the input space; (3) coreset affinity, which minimizes interference between selected samples and previously acquired knowledge, thereby promoting the stability across different tasks.

The gradient-based idea is also adopted by several bilevel formulations for coreset selection. For instance, Borsos et al. (2020) introduce a nested cardinality-constrained optimization model via the bilevel framework. Recall that we use $w$ to denote the weight vector of the given dataset $X = \{x_i\}_{i=1}^n$, $L(\theta; X, w) = \sum_{i=1}^n w_i \ell(x_i, y_i; \theta)$ the weighted empirical risk over $X$ for a model with the parameter vector $\theta \in \Theta$, and $L(\theta; X)$ the case where $w_i = 1$ for all $i$. Then, the bilevel formulation is

$$\hat{w} \in \operatorname*{argmin}_{w \in \mathbb{R}_+^n, \|w\|_0 \leq m} L(\theta^*(w); X) \quad \text{s.t.} \quad \theta^*(w) \in \operatorname*{argmin}_{\theta \in \Theta} L(\theta; X, w). \tag{20}$$

The solution of (20) can be obtained through the greedy forward selection method "Matching Pursuit (MP)" (Locatello et al., 2017). Hao et al. (2023) introduce another bilevel formulation which seeks to learn a probability distribution over a low-dimensional manifold by incorporating a smoothed top-K loss as the regularizer. Another method closely related to bilevel optimization is "Gradient Coreset Replay (GCR)" (Tiwari et al., 2022). In this approach, a coreset is constructed and maintained to approximate the aggregate gradient over all previously observed data items.

There also exist several other gradient-based approaches that focus on diversity and uncertainty. For example, Bang et al. (2021) highlight the importance of maintaining diversity within the episodic memory, since a memory buffer dominated by redundant or highly similar samples fails to provide sufficient coverage of the data distribution, which in turn limits the model's ability to retain previously learned knowledge. Then they propose "Rainbow Memory (RM)", a strategy to enhance diversity in continual learning.

### 3.3 Semi/Self-Supervised Learning

In this section, we introduce the coreset methods for two important scenarios of deep learning, *self-supervised learning* and *semi-supervised learning*. In general, self-supervised learning (Gui et al., 2024) utilizes only un-

labeled data, while semi-supervised learning (Yang et al., 2022) combines a small amount of labeled data with a much larger pool of unlabeled data. In particular, self-supervised learning has been a rapidly developing area in recent years, especially in its applications to large language models and multimodal models (Vaswani et al., 2017; Akbari et al., 2021). A representative self-supervised learning technique is *contrastive learning* (Chen et al., 2020; Tian et al., 2020). A prominent example is CLIP (Contrastive Language-Image Pretraining) (Radford et al., 2021), which extends contrastive learning to align images and texts in a shared embedding space, and has become a cornerstone framework for modern vision-language applications. The primary advantage of semi/self-supervised learning methods is their ability to learn without extensive labeled data. Nevertheless, a significant drawback is their demand for large-scale datasets and substantial computational resources. For alleviating this issue, coreset techniques have attracted a great amount of attention in recent years. These techniques contribute to improved computational efficiency (Killamsetty et al., 2021c; Dong et al., 2024), minimizing data redundancy (Wang et al., 2024b), and boosting robustness to distributional shifts (Xu et al., 2023).

Similar to the other scenarios discussed earlier, coreset selection in this setting is typically formalized as an optimization problem that aims at minimizing the performance gap between the models trained on the full dataset and a selected subset. Roughly speaking, the coreset should be as informative as the full unlabeled data in terms of the learned representations (Joshi et al., 2024; Wang et al., 2024b) or training dynamics (Killamsetty et al., 2021c; Dong et al., 2024; Kim et al., 2023a). Since finding the exact optimal subset is NP-hard in general cases, most practical approaches rely on approximate or heuristic algorithms. We categorize existing approaches by the primary proxy used to assess "data quality": **loss-based** (Wang et al., 2024b; Zhang et al., 2022), **gradient-based** (Killamsetty et al., 2021c; Dong et al., 2024) and **coverage-based** (Li et al., 2022; Fang et al., 2024; Zheng et al., 2025; Mayilvahanan et al., 2024; Nguyen et al., 2022; Li et al., 2024d) methods. A summary of several representative methods from each category is presented in Table 3.

### 3.3.1 Loss-Based Methods

Loss-based methods exploit the training loss or related signals to identify informative samples. The key intuition is that samples which incur higher loss values or provide stronger signals are more valuable for representation learning.

A typical example is the "s-CLIP-Loss" metric, proposed by Wang et al. (2024b). It generalizes the original CLIP training loss by incorporating a batchwise normalization term. This modification penalizes redundant samples whose image-text embeddings exhibit high similarity with incorrect text embeddings, thereby prioritizing specific and informative data samples. From a geometric perspective, s-CLIP-Loss refines the selection criterion by adjusting cosine similarity with a log-sum-exp aggregation of negative-pair similarities, acting as a soft redundancy-aware regularizer in the embedding space.

Another related concept is the dynamic leveraging of challenging samples, as implemented in the method called "hard negative mining" (Kalantidis et al., 2020; Zhang et al., 2022), which has been a widely used strategy in contrastive learning. Hard negatives are samples (or pairs) that the current model finds highly similar to the query sample despite being non-matching; in other words, these samples usually can provide a strong training signal to our coreset construction. Mining these hard negatives constitutes a form of dynamic data selection: at each training stage, the model selects the most informative pairs or samples to guide the learning process.

### 3.3.2 Gradient-Based Methods

Gradient-based methods aim to construct subsets that preserve the gradient statistics of the full dataset, so that optimization on the subset closely approximates the learning trajectory of the original training process. In this sense, the selected coreset captures the *training dynamics* of the model.

A well-studied approach is **gradient matching**. In Section 2.1, we have already introduced the idea of gradient matching for coreset construction. Roughly speaking, the gradient matching method tries to select a subset whose induced gradient could approximately represent those computed from the full dataset, thereby

Table 3: **Representative coreset selection methods in semi/self-supervised learning.** The "Availability" column indicates the public accessibility of the corresponding code or implementation: "official" refers to code released by the original authors, and "N.A." means no publicly available code has been found (to the best of our knowledge).

| Type | Methods | Highlights | Availability |
|---|---|---|---|
| Loss-Based | s-CLIP-Loss (Wang et al., 2024b) | Generalizing CLIP training loss with batchwise normalization to penalize redundant samples. | official |
| | Hard negative mining (Kalantidis et al., 2020; Zhang et al., 2022) | Dynamically selecting non-matching but highly similar samples (hard negatives) to provide strong contrastive signals. | official |
| Gradient-Based | RETRIEVE (Killamsetty et al., 2021c) | Selecting unlabeled samples with gradients aligning with labeled data for faster convergence. | official |
| | SkMM (Dong et al., 2024) | Integrating gradient-based dimensionality reduction with moment alignment for post-SSL fine-tuning. | official |
| | *Distribution Matching* | | |
| | ClipCov (Joshi et al., 2024) | Selecting image-caption pairs that preserve the cross-modal covariance matrix, enabling data-efficient multimodal contrastive pretraining. | official |
| | NormSim (Wang et al., 2024b) | Evaluating p-norm similarity between candidate samples and the target task distributions. | official |
| | *Diversity and Coverage* | | |
| Coverage-Based | SAS (Joshi & Mirzasoleiman, 2023) | Identifying examples preserving alignment and class center divergence by minimizing augmentation distance. | official |
| | ELFS (Zheng et al., 2025) | Using deep clustering for pseudo-labels and removing both simple and challenging samples. | official |
| | Mayilvahanan et al. (2024) | Pruning training data in CLIP embedding space using perceptual similarity as criterion. | official |
| | DeCLIP (Li et al., 2022) | Utilizing multi-view consistency and nearest-neighbor supervision to extract richer training signals. | official |
| | Nguyen et al. (2022) | Filtering datasets using pretrained models to enhance model robustness. | official |
| | Li et al. (2024d) | Focusing on high-quality subsets for contrastive pretraining under computational constraints. | N.A. |
| | Santurkar et al. (2023) | Showing high-quality textual captions provide superior transferability compared to larger image-only datasets. | N.A. |
| | Fang et al. (2024), Maini et al. (2024) | Identifying low-quality samples by quantifying drops in CLIP similarity scores on validation sets. | N.A. official |
| | SimCore (Kim et al., 2023a) | Coreset for fine-grained visual recognition under limited labels and abundant unlabeled open-set data. | official |

maintaining similar training dynamics. As mentioned before, RETRIEVE (Killamsetty et al., 2021c) applies this idea to improve both efficiency and robustness for semi-supervised learning.

Another important challenge in semi/self-supervised learning pipelines is how to select informative data for efficient *fine-tuning* after semi/self-supervised pretraining. Such fine-tuning is commonly required in downstream tasks, including linear probing or head tuning for classification, text-image retrieval, and lightweight adaptation for detection or segmentation, particularly under limited computational resources. The recent method "Sketchy Moment Matching (SkMM)" (Dong et al., 2024) addresses this problem by controlling the variance-bias trade-off that arises in high-dimensional fine-tuning. It first applies gradient sketching to project the full dataset's gradients onto a low-dimensional subspace that retains essential information of the model. Within this subspace, it then performs moment matching to select a coreset whose sketched gradient moments closely approximate those of the full dataset. This procedure is label-agnostic and thus it is well suited for resource-constrained fine-tuning after unsupervised pretraining.

### 3.3.3 Coverage-Based Methods

In contrast to the methods that focus on training dynamics, coverage-based methods construct coresets that preserve the geometry or distribution of the dataset in the representation space. Their core principle is that a good coreset should preserve the geometric or statistical properties of the full dataset's representation, such as its distribution, diversity, and coverage.

**Distribution matching.** A large part of coverage-based approaches follow the idea of distribution matching, which is previously introduced in Section 2.4. In the context of semi/self-supervised learning, a notable example is "ClipCov" (Joshi et al., 2024), a theoretically grounded coreset selection method tailored for CLIP (Radford et al., 2021). Unlike prior methods developed for supervised or unimodal contrastive learning, ClipCov specifically addresses the multimodal nature of CLIP by selecting image-caption pairs that can well preserve the cross-modal covariance structure. This distributional alignment is crucial for maintaining CLIP's generalization performance.

When the information of downstream tasks is available, the focus of distribution matching can align with the target distribution. A prime example of this approach is the "NormSim" method proposed by Wang et al. (2024b). This method guides the selection of semantically aligned subsets by evaluating the p-norm similarity between candidate samples and the target task distribution in the visual embedding space. This forward-looking selection strategy can yield a tighter coupling between the pretraining and fine-tuning stages. Specifically, it ensures that the coreset is constructed not only to preserve information from the original dataset, but also to achieve promising performance on specific downstream applications.

**Diversity and coverage.** Similar to the coreset construction methods introduced in Section 3.1.2 for active learning, there also exists a large body of work on semi/self-supervised learning that focuses on selecting a subset that maximally covers the data's semantic space. An example is from Joshi & Mirzasoleiman (2023), who introduce "Subsets that maximize Augmentation Similarity (SAS)", a method that identifies examples with preserving both augmentation alignment and class-center divergence among the input data. Similar to the coreset idea introduced in Section 2.1, this method formulates the subset selection as a submodular optimization problem. Furthermore, it provides a theoretical guarantee that the alignment and divergence measures computed on the selected subset uniformly approximate those computed on the full dataset, thereby ensuring that representations learned from the subset achieve comparable downstream generalization performance. The principle of maximizing coverage is also used in "SimCore" (Kim et al., 2023a), a coreset-based framework for fine-grained visual recognition under limited labels and abundant unlabeled open-set data. SimCore selects a subset from the open-set by maximizing its semantic similarity to a small target set, which is also formulated as a submodular optimization problem. The selection is performed in a learned representation space and is compatible with a variety of self-supervised methods, including the well-known approaches such as SimCLR (Chen et al., 2020), BYOL (Grill et al., 2020), DINO (Caron et al., 2021), and MAE (He et al., 2022). By sampling from the embedding space, SimCore ensures that the selected examples are both semantically relevant and diverse, which is particularly effective for adapting large-scale unlabeled data to specialized downstream tasks.

Other heuristics for achieving sufficient coverage in the representation space include *clustering-based selection* (e.g., $k$-center or $k$-means on pretrained embeddings) to pick cluster prototypes that span high-density regions (Zheng et al., 2025), and *redundancy-aware pruning*, which removes overly similar samples to reduce representation overlap (Mayilvahanan et al., 2024). Zheng et al. (2025) introduce "Effective Label-Free Coreset Selection (ELFS)", a method for coreset selection without relying on ground-truth labels. ELFS utilizes deep clustering technique (Adaloglou et al., 2023) to infer pseudo-labels and approximates difficulty scores based on training dynamics such as the model's behavior across epochs measured by indicators including the Area Under the Margin (AUM) (Pleiss et al., 2020), forgetting events (Toneva et al., 2019), and early-epoch loss (EL2N) (Paul et al., 2021). To mitigate noise and distributional shifts introduced by pseudo-labels, it adopts a double-ended pruning strategy that prunes both trivially easy and overly hard samples, thereby improving the representativeness and overall utility of the selected coreset.

Mayilvahanan et al. (2024) conduct a systematic study of pruning in CLIP's embedding space to examine whether its strong out-of-distribution (OOD) generalization mainly stems from high train and test similarity. They define a *perceptual similarity* metric, computed as the cosine similarity between CLIP image

embeddings, to quantify visual closeness in both content and style, and progressively remove training samples that are overly similar to specific test distributions. Even after removing these high-similarity samples, CLIP's zero-shot accuracy on OOD benchmarks such as ImageNet-Sketch (Wang et al., 2019) and ImageNet-R (Hendrycks et al., 2021) decreases only moderately and remains far higher than that of models trained solely on ImageNet (Deng et al., 2009). They further show that a 100M subset obtained through far-pruning achieves nearly the same accuracy as the full 400M dataset, suggesting that large-scale diversity and cross-modal supervision, rather than sample-level overlap, are the main factors driving CLIP's generalization. These findings offer useful insights for designing data-efficient pretraining and diversity-preserving coreset strategies.

The aforementioned approaches, also broadly categorized as "data filtering" or "data curation", highlight a growing consensus: data quality is often more critical than sheer quantity. Li et al. (2022) introduce "DeCLIP", a framework designed to address redundancy in large-scale image-text datasets. By integrating additional self-supervised objectives, multi-view consistency, and nearest neighbor supervision, DeCLIP extracts richer training signals from smaller, curated subsets. Such implicit coreset selection mechanisms leverage model-based filtering strategies to curate high-quality subsets. This strategy embodies the coreset principle of achieving high learning efficacy using minimal yet informative data. Similarly, a number of recent works also underscore the importance of subset quality over sheer dataset size (Nguyen et al., 2022; Santurkar et al., 2023; Li et al., 2024d; Fang et al., 2024; Maini et al., 2024; Joshi et al., 2024).

### 3.4 Summary

We have introduced the coreset selection methods for enhancing data efficiency in three scenarios with limited resource. Generally speaking, the coreset methods developed for these scenarios can be divided into four categories, that is, loss-based (see Section 3.1.1 and 3.3.1), gradient-based (see Section 3.2.2 and 3.3.2), coverage-based (see Section 3.1.2, 3.2.1 and 3.3.3), and hybrid methods (see Section 3.1.3). Loss-based methods aim to select examples that are likely to incur higher loss values and prioritize those about which the model is most uncertain (Mayer & Timofte, 2020; Kim et al., 2021b; Gao et al., 2020; Parvaneh et al., 2022). However, relying on uncertainty alone risks selecting redundant or similar examples that contain highly overlapping information (Citovsky et al., 2021; Kim & Shin, 2022; Ash et al., 2020; Cho et al., 2024). Gradient-based methods primarily select examples that could preserve the gradient statistics of the full dataset (Killamsetty et al., 2021c). Coverage-based methods, on the other hand, emphasize diversity by selecting representative and varied examples that enable broad coverage of the input space (Yehuda et al., 2022; Bae et al., 2024; Joshi & Mirzasoleiman, 2023; Kim et al., 2023a). Therefore, these categories prioritize examples that are diverse and dissimilar (Aljundi et al., 2019; Yoon et al., 2022). But focusing exclusively on diversity leads to annotating examples that are trivial or already well learned (Citovsky et al., 2021; Kim & Shin, 2022; Ash et al., 2020). Hybrid methods are proposed to mitigate these shortcomings by simultaneously considering both uncertainty and diversity (Citovsky et al., 2021; Kim & Shin, 2022; Ash et al., 2020; Cho et al., 2024). The goal is to select examples that not only provide novel information but also adequately represent the underlying data distribution. As a result, hybrid methods could offer a more effective trade-off between uncertainty and diversity.

## 4 Coresets for Large Language Models

Large language models (LLMs), such as DeepSeek-v3 (Liu et al., 2024), GPT-4 (Achiam et al., 2023), and LLaMa (Touvron et al., 2023), exhibit remarkable proficiency across a broad spectrum of language understanding and generation tasks. In their training procedures, LLMs typically follow two key stages (Ouyang et al., 2022): *pretraining* on large scale corpora and *fine-tuning* on instruction following datasets. Pretraining constitutes a fundamental stage in the development of LLMs, a stage that equips the models with core knowledge and capabilities. LLMs acquire a deep understanding of language syntax, world knowledge, and reasoning abilities through self-supervised learning on extensive textual corpora. This stage lays the groundwork for fine-tuning, a process in which models are further trained on a carefully annotated set of (instruction, response) pairs so that their capabilities are enhanced and their controllability is improved.

Similar with the studies on semi/self-supervised learning in Section 3.3, recent studies suggest that the quality of training data also plays a more critical role than its quantity for LLMs (Zhou et al., 2023). Consequently, a growing body of works focus on selecting high-quality subsets from large-scale pretraining or fine-tuning datasets. In this section, we provide a comprehensive overview of recent coreset-based data selection techniques for the pretraining and fine-tuning stages of LLMs.

## 4.1 Coresets for Pretraining

The pretraining of a large language model is significantly influenced by the characteristics of the training corpus (Longpre et al., 2024; Chowdhery et al., 2023). Existing research shows that model performance can be substantially improved through the careful curation of high-quality data (Sachdeva et al., 2024; Wettig et al., 2024). In particular, data quality, diversity, and coverage are widely recognized as the key factors in enhancing the training efficiency and generalization ability of LLMs (Cheng et al., 2024; Chowdhery et al., 2023; Touvron et al., 2023).

To improve data quality, rule-based filtering techniques have been widely adopted (Weber et al., 2024; Touvron et al., 2023; Laurençon et al., 2022; Penedo et al., 2023). These methods rely on manually designed heuristics, such as removing terminal symbols, detecting repetitive sentences, or enforcing length constraints, to filter out low-quality samples from the training corpus. While being effective in eliminating superficial noise, such approaches lack the capacity to capture semantic-level information, which is critical for fine-grained and content-aware data selection.

Several studies have been proposed to address the above limitation. For example, Wang et al. (2023a) introduce a method called "Influential Subset Selection (ISS)", which explicitly leverages knowledge of the end task to guide pretraining data selection. Specifically, ISS selects the samples that have the greatest positive influence on the performance of the end task. Furthermore, the authors design a gradient-matching-based influence estimation method, which drastically reduces the time required to estimate influence. Using only 0.45% of the data and achieving a computational cost that is three orders of magnitude lower, ISS outperformed pretrained models (e.g., RoBERTa) on eight datasets covering four domains. Complementary to this line of work, recent research (Thrush et al., 2025) demonstrate that high-quality pretraining data can be selected by exploiting the correlation between language model perplexity and downstream task performance. Specifically, the "perplexity" is computed as the normalized sequence negative log-likelihood. The authors propose to select the domains of pretrained corpus with largest correlations between perplexity and target benchmark scores. Such correlation can be estimated by the output logits computed from a sample of 90 LLMs taken from the "Hugging Face Open LLM Leaderboard"[1], without the need for additional LLM training. Marion et al. (2023) perform a rigorous comparison between the simple data quality estimator of perplexity and more computationally intensive estimates of the Error L2-Norm (Paul et al., 2021) and Memorization (Biderman et al., 2023), and find that the simple perplexity-based method outperforms the more computationally expensive approaches. Lin et al. (2024b) introduce a novel language model named "RHO-1", which is different from the conventional LLMs who predict every next token in the corpus. Instead, RHO-1 focuses on learning from tokens aligned with the target distribution. This is achieved by assigning scores to tokens using a reference model and applying a loss function that prioritizes higher-scoring (i.e., more informative or relevant) tokens during training. Similar to the coreset idea introduced in Section 2.2, Xie et al. (2023) propose "Data Selection with Importance Resampling (DSIR)", an efficient and scalable framework that estimates importance weights in a reduced feature space. The data items are then selected via importance resampling based on these weights. To ensure efficiency, the authors instantiate DSIR using hashed n-gram features, enabling the selection of 100 million documents from the full Pile dataset within 4.5 hours on 4 Titan RTX GPUs.

Another line of research adopts a discriminator-based approach for data selection (Li et al., 2024b; Soldaini et al., 2024; Brown et al., 2020; Du et al., 2022). Rather than estimating sample importance or influence, these methods compare candidate data to predefined high-quality corpora, such as Wikipedia or instruction-tuning datasets, using trained discriminators to identify the samples that resemble the reference distribution. But such approaches heavily depend on the qualities of the reference datasets. So, a number of recent

---

[1] https://huggingface.co/spaces/open-llm-leaderboard/open_llm_leaderboard#/

studies (Penedo et al., 2023; Wettig et al., 2024; Sachdeva et al., 2024) turn to find different solutions, e.g., leveraging existing large language models as the data evaluators. By employing carefully designed prompts to assess data quality from various dimensions (e.g., factuality, coherence, and task relevance), these methods enable more nuanced and flexible filtering criterion for data selection. This LLM-assisted evaluation approach offers a promising alternative to discriminator-based approach, facilitating more robust and scalable pretraining data selection.

Beyond the filtering and discriminator-based methods, several recent works focus on optimizing the distributional characteristics of the training corpus through clustering (Abbas et al., 2023; Shao et al., 2024). These approaches aim to reduce redundancy while preserving semantic diversity. For example, Abbas et al. (2023) propose the method "SemDeDup", which leverages embeddings from pretrained models to detect and eliminate semantic duplicates, which are pairs of examples that are highly similar in meaning but not necessarily identical. Shao et al. (2024) develop the technique "ClusterClip" to balance the text distribution of training data for better model training. Specifically, ClusterClip utilizes clustering to capture the data distribution of the training set and balances the common and rare samples during the training procedure.

## 4.2 Coresets for Fine-Tuning

Fine-tuning plays a critical role in aligning LLMs with human instructions. By training on (instruction, output) pairs, fine-tuning helps bridge the gap between pretrained models and diverse human intents. Specifically, it enables LLMs to generate outputs that better reflect human preferences, thereby improving their controllability and safety. Moreover, fine-tuning can facilitate the adaptation to specific domains or acquisition of task-specific knowledge, without requiring substantial computational resources or architectural modifications.

Recent study (Zhou et al., 2023) demonstrates that fine-tuning with only 1,000 carefully curated examples can yield strong performance comparable to the models trained on datasets that are several orders of magnitude larger. Very recently, several model-dependent data selection methods for efficient fine-tuning have been proposed. Joaquin et al. (2024) provide the method "In2Core", which leverages internal model gradients to quantify the influence of each training instance. Nguyen et al. (2025b) introduce "CoLM", a memory-efficient training method for LLMs, where it identifies compact mini-batches to approximate the gradients. Similar to the coreset idea introduced in Section 2.1, they model the problem as a submodular maximization problem with leveraging pairwise gradient similarities. Zhang et al. (2025) introduce "STAFF", an efficient coreset selection strategy designed for task-specific fine-tuning of LLMs. The method first employs a smaller model from the same architecture family to estimate data importance scores, which are then verified and refined in the target LLM. This two-stage process enables accurate identification of high-impact data regions while preserving overall diversity. Yang et al. (2024) propose "SmallToLarge (S2L)", a data selection method that first trains a small reference model, clusters examples based on their loss trajectories, and then samples data from these clusters to guide the training of larger models.

## 4.3 Summary

This section provides an overview of recent advances in coreset-based data selection methods for both the pretraining (in Section 4.1) and fine-tuning (in Section 4.2) stages of large language models. During pretraining, high-quality data selection plays a crucial role in improving model efficiency and generalization. Existing approaches include rule-based filtering (Weber et al., 2024; Touvron et al., 2023; Laurençon et al., 2022; Penedo et al., 2023), discriminator-based selection (Li et al., 2024b; Soldaini et al., 2024; Brown et al., 2020; Du et al., 2022) using pretrained models as evaluators, and distributional optimization techniques that better align training corpora with target distributions (Abbas et al., 2023; Shao et al., 2024). Representative methods such as DSIR (Xie et al., 2023) enable efficient large-scale selection, while ISS (Wang et al., 2023a) identifies influential subsets through gradient-based influence estimation. Additional techniques such as clustering (e.g., SemDedup (Abbas et al., 2023), ClusterClip (Shao et al., 2024)) further enhance data quality by reducing redundancy and maintaining diversity.

For fine-tuning, the focus shifts toward selecting informative instruction-response pairs that can maximize downstream performance while reducing annotation and computational cost. Methods such as

In2Core (Joaquin et al., 2024), CoLM (Nguyen et al., 2025b), STAFF (Zhang et al., 2025), and S2L (Yang et al., 2024) leverage model-internal signals or smaller proxy models to estimate data importance, cluster examples, and identify high-impact regions of the data space. These techniques enable models to achieve competitive performance with only a small fraction of the original data.

## 5    Challenges and Future Works

This survey reviewed the coreset methods in deep learning, which enhance data efficiency and preserve performance by selecting small, representative data subsets. We outlined their foundational principles and roles: reducing computational cost for large-scale training (as introduced in Section 2) and improving data utilization in resource-constrained scenarios (as introduced in Section 3) such as active learning, continual learning, and semi/self-supervised learning. We also explored the growing relevance of coreset techniques in large language models in Section 4. Despite the significant advances in coreset techniques, several critical challenges and promising future directions still remain in this field. At the end of this article, we highlight three key open issues that are pivotal to advancing both the theory and practical applications of coresets.

### 5.1    Dynamic and Adaptive Coreset Selection

Traditional coreset techniques often assume the given datasets are static. However, real-world scenarios frequently involve dynamic data streams or continual learning settings, where data distributions evolve over time. Although coresets have been successfully designed for fully dynamic data settings in models like SVMs and linear regression (Wang et al., 2021), extending these capabilities to deep neural networks remains a significant challenge. In particular, designing algorithms that can adaptively update coresets in response to streaming data or distributional shifts is still an open problem (Aljundi et al., 2019; Buzzega et al., 2020).

Recent findings reveal that in large-scale pretraining scenarios, especially with contrastive learning models such as CLIP (Radford et al., 2021), the utility of high-quality data can decay rapidly with repetition (Goyal et al., 2024). This highlights a critical limitation of static or heuristic-based coreset strategies, which often ignore the interplay between data utility, training repetitions, and compute budgets (such as available GPU hours or memory size).

Future coreset algorithms should not only track data distributional shifts but also estimate and model repetition-induced utility decay across heterogeneous data pools. Recent advances in scaling-law analysis have begun to formalize this idea by explicitly modeling how the marginal utility of data decays with repeated exposures, and how the optimal data-filtering strength depends on available compute resources. In particular, Goyal et al. (2024) demonstrate that data curation cannot be compute-agnostic: they derive a scaling law that predicts performance across heterogeneous datasets by jointly modeling data quality, repetition, and compute budget. This framework provides a principled foundation for predicting the performance of different coreset configurations under varying compute budgets.

Moreover, adapting coreset strategies to the model architecture and training dynamics (e.g., ViT vs. CNN, contrastive vs. generative objectives) is becoming increasingly important (Li et al., 2024d). For instance, large vision transformers exhibit greater performance gains from data diversity than smaller models (Li et al., 2024d). This effect arises because larger models possess higher representational capacity and are often trained with fine-tuning techniques that enable them to leverage more informative samples. In contrast, smaller models tend to saturate earlier and gain less from additional diversity. Consequently, the coreset construction should be considered jointly with model scaling to ensure that the selected subset matches the model's capacity and fine-tuning methods.

Furthermore, the development of automated coreset construction represents a promising direction. A practical challenge in deploying coreset methods lies in determining the appropriate coreset size and selection criteria, which currently depend on manual tuning or domain-specific heuristics. This tuning often requires balancing trade-offs among accuracy, robustness, fairness, and computational efficiency. Automated coreset construction, potentially leveraging AutoML or meta-learning paradigms, could more efficiently discover minimal coresets tailored to specific objectives. Early efforts in this direction, such as

RETRIEVE (Killamsetty et al., 2021c), demonstrate that meta-learned selection strategies can adaptively construct coresets aligned with downstream task requirements.

## 5.2 Balancing Fidelity and Diversity

Existing coreset methods frequently depend on single, isolated selection criteria, such as maximizing diversity or prioritizing regions of high data density, potentially leading to suboptimal trade-offs between fidelity and diversity. For instance, the methods emphasizing dense, high-quality regions risk neglecting rare but critical modes, thus reducing generalization capabilities and diversity in generative tasks. On the other hand, the methods that solely optimize for diversity may include noisy or irrelevant data points, diminishing the overall performance.

Recent findings further highlight that diversity and fidelity cannot be treated as independent or uniformly beneficial dimensions. In large-scale multimodal pretraining, data from heterogeneous sources often exhibits complementary but non-additive robustness properties (Nguyen et al., 2022). Moreover, simply increasing dataset diversity without assessing semantic quality or target-task relevance can worsen performance under distribution shifts. This underscores the importance of not only balancing but also contextualizing fidelity-diversity trade-offs based on source characteristics and alignment with downstream tasks.

To address these challenges, recent studies propose more metric-informed data selection strategies for multi-modal pretraining. For instance, the classical CLIPScore (Hessel et al., 2021) measures the cosine similarity between visual and textual embeddings from CLIP, serving as a simple proxy for sample quality. However, it often overestimates generic or weakly aligned pairs. To mitigate this, s-CLIP-Loss (Wang et al., 2024b) refines quality estimation by incorporating a contrastive-pair normalization term inspired by the original CLIP training loss, yielding a more faithful assessment of multimodal consistency. In parallel, NormSim (Wang et al., 2024b) defines a vision-only p-norm similarity between pretraining data and target domains, allowing models to prioritize samples that are semantically relevant to downstream tasks. Together, these complementary metrics suggest that data fidelity and task relevance should be optimized jointly but with different priorities, that is, rare yet informative examples are retained, while redundant or mismatched samples are pruned more aggressively.

Future research should therefore pursue hybrid coreset frameworks that incorporate both structural diversity and semantically-informed fidelity signals. Such systems could dynamically tune sampling strategies using proxy metrics like task-targeted NormSim, or decay-based models of utility (Goyal et al., 2024), allowing coresets to remain compact yet highly expressive across different tasks and training phases. This direction holds promise for building coreset selection methods that are robust, efficient, and capable of generalizing across diverse tasks (Cho et al., 2024; Citovsky et al., 2021).

## 5.3 Ethics and Privacy

As coreset selection becomes increasingly prevalent, it raises significant concerns regarding algorithmic fairness and data privacy. The selection process itself, while designed for efficiency, can inadvertently amplify systemic biases encoded in the original data. For example, selection criteria may disproportionately favor majority groups or samples that align with spurious correlations, thereby degrading the group robustness of downstream models (Dharmasiri et al., 2025). Moreover, the resulting coreset, being a concentrated and highly representative summary, could inadvertently leak sensitive information or make individuals more identifiable, posing a privacy risk. While early explorations into private coresets have been studied for tasks like clustering (Feldman et al., 2009; 2017), ensuring these properties in broader applications remains a challenge. Thus, developing coreset selection frameworks that integrate formal guarantees, such as differential privacy principles and explicit fairness criteria, represents a vital and promising research direction for trustworthy machine learning (Fioretto et al., 2022).

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
