# OpenReview forum: "Survey on Coresets for Deep Learning: Methods and Applications"
_TMLR — Rejected by TMLR_

### Review · Reviewer_xZT8 · 2026-01-09

**Summary Of Contributions:**

This paper provides a comprehensive survey of the coresets methods in the last 20 years of deep learning research.
- It contains a broad range of the coresets methods and proposes a clear taxonomy for classifying them into greedy selection, importance sampling, filtering, and distribution matching groups.
- This paper organizes the coreset application into two primary scenarios: one is for efficient model training, and another is for improving data utilization under resource constraints.
- This paper extends the survey to the popular area of Large Language Models, and summarizes how the coreset method is applied to LLM pretraining and fine-tuning.

**Audience:**

Yes

**Audience Explanation:**

This paper fills a gap left by previous surveys that focused primarily on classical machine learning and provides a systematic survey of the core of deep learning. Researchers interested in scalable training in resource-constrained data scenarios may gain valuable insights from this paper.

**Broader Impact Concerns:**

N/A, this paper is a survey of existing methods and does not introduce new things with ethical concerns.

**Claims And Evidence:**

Yes

**Claims Explanation:**

The survey paper cites a large amount of technical papers to support their built taxonomy, application, and discussion about the key challenges and promising directions in this field.

**Requested Changes:**

The paper currently focuses on qualitative analysis and comparison. To better evaluate different methods, it is suggested to include quantitative comparison results.

- The experiments are conducted on a unified setting (e.g., for image classification and more tasks) to provide a fair benchmark.

- In addition to the main performance table, it would be beneficial to include a discussion on complexity. Calculating the computational overhead of coreset construction approaches would make the analysis more comprehensive.

---

> ### Author Response · Authors · 2026-02-04
>
> **R1: The experiments are conducted on a unified setting (e.g., for image classification and more tasks) to provide a fair benchmark.**
>
> **A1:** Thank you for the suggestion. We agree that quantitative comparisons would strengthen the paper beyond the current qualitative analysis. In the revised version, we will add a unified quantitative comparison for representative coreset methods **across several key tasks covered in our survey**, including **active learning, continual learning, and semi/self-supervised learning**. For each task, we will adopt a common experimental protocol (consistent datasets, models, training setting and multiple coreset budgets) and report consolidated results in a dedicated table to enable fair, task-specific benchmarking.
>
> **R2: In addition to the main performance table, it would be beneficial to include a discussion on complexity. Calculating the computational overhead of coreset construction approaches would make the analysis more comprehensive.**
>
> **A2:** We fully agree that, beyond predictive performance, the computational cost of constructing coresets is crucial for a fair assessment. In the revised version, we will add a dedicated complexity and overhead discussion and a summary table reporting the computational overhead of representative coreset construction approaches, including (when available) their time and memory costs (e.g., asymptotic complexity and normalized wall-clock runtime) and the resulting end-to-end training cost trade-offs.

---

> > ### Comment · Reviewer_xZT8 · 2026-02-04
> >
> > Thanks for the reply.
> >
> > I couldn't find the revised version. Did you forget to upload it? Please highlight the additional contents in the submission PDF. Thanks

---

### Review · Reviewer_RAhd · 2026-01-12

**Summary Of Contributions:**

The authors present a thorough summary of coresets: the history, an overview of methods, and the remaining future work. The overview of methods is broken down into problem patterns (broadly: greedy selection via utility functions, importance sampling via scoring, filtering, and distribution matching), data utilization (active learning, continual learning, semi-supervised learning), and applications in LLMs (pretraining, fine-tuning). The remaining works cover the up-and-coming problem of dynamic data selection, diversity, and ethical privacy. The survey is in depth and generally well written.

**Audience:**

Yes

**Audience Explanation:**

As someone interested in coresets, this survey helped clear up my own mental map of coresets. They also provide summaries at the end of each section which helps tie the current section and next section together.

**Broader Impact Concerns:**

There are no ethical concerns with their survey, but they do cover the ethical considerations of coreset selection, which seems to address this.

**Claims And Evidence:**

Yes

**Claims Explanation:**

They thoroughly explain each claim backed with plentiful citations and evidence. They clearly lay out the mathematical formulations and concepts.

**Requested Changes:**

I think the survey is very well organized and fleshed out already. I only have suggestions that would strengthen the work:
- In Section 2.2, the authors cover importance sampling wrt neural networks, GANs, and training dynamics quite extensively. In my opinion, it seems to diverge from the main topic of discussion.
- A small discussion in Section 3 about few-shot learning or data augmentation could help tie the resource-constrained environments overview together.
- Section 4 seems a bit underwhelming given the amount of data efficiency works with LLMs. Perhaps, another literature review could strengthen this particular section.

---

> ### Author Response · Authors · 2026-02-04
>
> **R1: In Section 2.2, the authors cover importance sampling wrt neural networks, GANs, and training dynamics quite extensively. In my opinion, it seems to diverge from the main topic of discussion.**
>
> **A1:** We thank the reviewer for this comment. The goal for mentioning neural networks, GANs, and training dynamics is to   provide the practical context for importance sampling in deep-learning coreset  methods. We will revise the presentation to make this connection more explicit and keep the discussion more focused, so that the scope of Section 2.2 is clearly aligned with the main topic of the survey.
>
> **R2: A small discussion in Section 3 about few-shot learning or data augmentation could help tie the resource-constrained environments overview together.**
>
> **A2:** That is a very helpful suggestion and aligns with the goal of Section 3. In the revision, we will add a brief discussion in Section 3 on **few-shot learning** and **data augmentation** as complementary strategies for resource-constrained environments, clarifying how they differ from coreset-based approaches and how they can be combined (e.g., using coresets to select data, and augmentation or few-shot techniques to further boost performance under tight budgets).
>
> **R3: Section 4 seems a bit underwhelming given the amount of data efficiency works with LLMs. Perhaps, another literature review could strengthen this particular section.**
>
> **A3:** Thank you for the suggestion. We agree that Section 4 can be strengthened to better reflect the breadth of data-efficiency research for LLMs. In the current version, Section 4 is organized into pretraining and fine-tuning. In the revised version, we will  expand the literature coverage in this section: we will update the pretraining part with additional recent works on data-efficient LLM pretraining, and we will refine the fine-tuning part into more fine-grained stages (instruction-tuning, alignment, in-context learning, and task-specific fine-tuning), each with a brief survey of representative methods. We will also add a summary table that positions representative works across stages.

---

> > ### Comment · Reviewer_RAhd · 2026-02-27
> > **No updated manuscript provided?**
> >
> > Hello, as the other reviewers have mentioned, the authors do not seem to have uploaded an updated manuscript. While I do maintain my recommendation of acceptance, I also acknowledge the concerns/questions/suggestions of other reviewers have not been addressed.

---

### Review · Reviewer_5Jna · 2026-01-21

**Summary Of Contributions:**

Coresets have become an important tool in deep learning, yet there is no clear, universally accepted definition of a coreset in the literature. This paper surveys how the notion of a “coreset” is used in modern deep learning pipelines. It adopts a broad perspective, viewing coresets as small, often weighted, subsets intended to approximate training on the full dataset. The paper argues that, in deep learning, the term has moved beyond its classical approximation-theoretic roots and now covers a wider range of practical approaches.

Strength:

1. Timely scope and topic relevance. The survey covers many recent methods. It explicitly includes LLM data selection for pretraining and fine-tuning, which increases practical relevance.

2. Clear paper-by-paper descriptions. Many methods are explained in an accessible way, and the survey provides useful background formulations.

3. Useful tables are provided.

Weakness:

1. The structure is use-case driven, but many methods are not tied to a single use case. This makes the overall partitioning feel overlapping. For example, Paul et al. (2021) appears in Sections 2.3, 3.3, and 4.1. What is the rationale for organizing the paper in a use-case-driven way?

2. Logical connections and distinctions across method categories are not fully developed. Some new methods in different categories are proposed to solve the flaws in previous methods, but the logical connections are not given.

3. Limited comparative analysis. The paper offers brief trade-off notes in the summaries; however, these comparisons are not systematic. The common parts of the methodology should be addressed more clearly to show the distinctions.

4. No benchmark or metric is provided to show what a good coreset is. There is no benchmark to compare the performance of different methods. I understand that benchmarking may be outside the scope of this paper; however, at least some useful metrics should be proposed to define a “good” versus “bad” coreset. In addition, some runtime analysis could be included to show the time complexity of different methods, to better illustrate the trade-off between performance and resource requirements.

**Audience:**

Yes

**Audience Explanation:**

This topic is highly relevant to TMLR readers working on data-efficient learning, training acceleration, and especially dataset selection/curation for large-scale models (including LLMs). The tables (Tables 1–3) and the inclusion of LLM-focused material make the paper potentially useful as a reference.

That said, to maximize impact for TMLR’s audience, the survey likely needs stronger synthesis, comparison, and guidance. Compared with AI-generated surveys, a paper that mainly compiles and summarizes existing work adds little value unless it offers something AI summaries usually cannot: a well-justified scope, a clear and unified way to organize the field, and a critical comparison of methods, including their trade-offs and failure cases.

**Broader Impact Concerns:**

NA.

**Claims And Evidence:**

Yes

**Claims Explanation:**

Overall, the manuscript does a solid job of accurately summarizing the definitions and core claims of many referenced papers, and it often anchors descriptions with citations and standard formulations.

However, the survey’s organizational claims are less convincing as written. The division into major sections/subsections is not always conceptually clean, and the paper itself acknowledges this overlap: it states the method categories are “not mutually exclusive”, and the Sec. 2.5 summary explicitly explains that the same construction principles recur across later sections.

In addition, while the paper includes brief trade-off commentary in summaries, it does not consistently provide comparative evidence or synthesized conclusions about relative strengths/weaknesses across method families. As a result, the survey can read more like a collection of method summaries than a unified “survey argument” with clear structure. It is more “bricks” than a finished building.

**Requested Changes:**

1. Clarify and operationalize the scope of “coreset” in this survey.

1.1 The paper explicitly says unified definitions are unrealistic and that there is “no strict boundary” with dataset distillation/curation. This is fine, but the survey should then provide clear inclusion criteria.

1.2 More comparison and discussion on the trade-offs between coresets and data distillation/curation should also be provided to better guide the audience on whether a coreset or data distillation/curation is what they need. More explanation and comparison are necessary, since there is no strict boundary between coresets and these two similar techniques in reality, and some methods in data distillation claim they are coreset methods. For example, in some literature on Wasserstein coresets [1][2], the authors claim they are coreset methods, but due to the nature of the Wasserstein distance and optimal transport, the “coresets” they generate are barycenters, which are usually synthetic small datasets rather than representative subsets of actual samples.

2. Provide better logical connections between method categories. For example, I noticed some methods that appeared in Section 2.3 reappeared in Section 3.3 (Paul et al., 2021; Toneva et al., 2019), some methods that appeared in Section 2.1 reappeared in Section 3.3 (Killamsetty et al., 2021c; Xu et al., 2023), and some methods that appeared in Section 2.1 reappeared in Section 3.2 (Borsos et al., 2020; Hao et al., 2023). If there are connections between these section definitions, the connections should be addressed, as well as the differences in definitions across sections.

3. Add systematic comparative discussion of advantages, disadvantages, and practical trade-offs. The summaries already contain brief trade-off notes, but the survey needs a more systematic treatment. If the target audience is researchers who develop new coreset methods, then it does not make sense for the survey to be organized in a use-case-driven way, since many methods are not tied to a single use case. If the target audience is people who want to use coreset methods in applications, then more comparisons between methods, performance discussions, and trade-offs should be given. I understand that a benchmark might be too much, but some metrics used to compare the performance of coreset methods should be discussed.

4. A few typos,

4.1 “uncertainy” in the active learning section (p.12).

4.2 In algorithm 1, when a set is unioned with an element, it should be $S \cup$ \{ $x$ \}, instead of $S\cup x$.


[1] Claici, S., Genevay, A., & Solomon, J. (2018). Wasserstein measure coresets. arXiv preprint arXiv:1805.07412.

[2] Yin, H., Qiu, Y., & Wang, X. (2025). Wasserstein coreset via sinkhorn loss. Transactions on Machine Learning Research.

---

> ### Author Response · Authors · 2026-02-04
>
> **R1: Clarify and operationalize the scope of “coreset” in this survey. Provide clear inclusion criteria. More comparison and discussion on the trade-offs between coresets and data distillation/curation should also be provided.**
>
> **A1:** We thank the reviewer for this important suggestion.
>
> In this survey, coreset methods are defined as those that select a small subset consisting of actual samples from the original dataset. This is reflected in the statement in the paragraph before formula (1) in Page 2: "The objective of coreset construction is to identify a small, weighted subset $S = \{ \hat{x} \}_{i=1}^k \subseteq X$". In contrast, some data distillation methods  generate synthetic data rather than selecting actual samples, and we do not include them in our survey.
>
> We appreciate the reviewer for mentioning the articles \[1\] and \[2\], and we will add these two references. Although these two methods do not directly select a subset from the input data, they select a subset from the input data's support. So we can recognize them as "coreset" methods within a broader range (but of course, they can be also regarded as data distillation methods from some sense).
>
>  We acknowledge that there is conceptual overlap between coresets and data curation. Data curation methods that select actual samples are treated as coreset methods in this survey, as noted in the final paragraph of Section 3.3, where we recognize that some introduced methods are “broadly categorized as data filtering or data curation.” Conversely, the data curation approaches that generate synthetic data are not considered as coresets.
>
> In the revised version, we will further clarify the scope of coresets and include more comparison and discussion between coresets, data distillation, and data curation. We thank the reviewer again for the suggestion.
>
> **R2: Provide better logical connections between method categories.**
>
> **A2**: We thank the reviewer for the thoughtful suggestion. We agree that the rationale and connections within our use-case-driven categorization could be presented more clearly.
>
> First, we recognize that that there are overlaps between method categories, as noted in the survey. But it is often impossible to create perfect and non-overlapping categories. We try to minimize their overlaps  while ensuring the categories reflect the primary objectives for which coreset methods are originally designed: "efficient training" and "improved data utilization". We believe this categorization aligns with how researchers design and practitioners deploy coreset techniques.
>
> Also, this categorization does not imply that a given method is restricted to a single use case. As the reviewer observes, some methods (such as Paul et al., 2021)  appear in multiple sections because they are sufficiently general and can be effectively applied across different scenarios.
>
> Finally, we include a separate section on coresets for LLMs since they are currently a major focus of the research community and there are many coreset methods designed for LLMs. We therefore believe it is beneficial to highlight this emerging direction explicitly.
>
> We agree with the reviewer that the rationale for such a use-case-driven categorization and logical connections and distinctions across method categories were not clearly discussed in the original manuscript. In the revised version, we will clarify the motivation behind the categorization and the reason why some methods appear across multiple sections. We thank the reviewer again for helping us improve the clarity and presentation of the survey.
>
> **R3: Add systematic comparative discussion of advantages... compare the performance of coreset methods should be discussed.**
>
> **A3:** Thank you for the suggestion. We agree that the survey would benefit from more systematic, quantitative comparisons. In the revised version, we will add section-aligned **quantitative experiments**: one in **Section 2** (coresets for efficient training), and one each in **Sections 3.1–3.3**, corresponding to active learning, continual learning, and semi/self-supervised learning, respectively. For each setting, we will evaluate representative methods under a **unified protocol** (consistent datasets/models/training settings and multiple coreset budgets) and summarize the results in tables, enabling clearer discussion of advantages, limitations, and practical trade-offs. We will also discuss several theoretical aspects, such as coreset construction time.
>
> **R4: A few typos.**
>
> **A4:** We thank the reviewer for pointing out these typos. In the revised version, we will correct them. We will also carefully proofread the manuscript for any remaining typographical errors.

---

> > ### Comment · Reviewer_5Jna · 2026-02-20
> > **Will there be updated manuscript provided?**
> >
> > I appreciate the detailed response.
> >
> > Now the ddl for submitting final recommendation has passed. If there is no updated manuscript provided by the end of this week, I will submit the final recommendation based on the original version.
> >
> > Thank you!

---

### Decision · Action_Editor_R2ha · 2026-03-01

**Recommendation:** Reject

**Audience:**

Yes

**Audience Explanation:**

This survey could be of interest to researchers working on coreset methods, data selection, and related areas.

**Claims And Evidence:**

No

**Claims Explanation:**

This paper provides a survey of coreset methods for deep learning learning, covering their historical development, overview of existing approaches, and open directions for future research. The survey draws on a large body of technical literature to support its proposed taxonomy, applications, and discussion of key challenges and promising directions in the field.

However, the reviewers provided several suggestions for improvement, and the authors indicated their willingness to revise the manuscript. But no revised version has been submitted. Given that the required revisions may go beyond minor edits and the majority of reviewers recommend rejection, I concur with their assessment.

**Resubmission Of Major Revision:**

The authors may consider submitting a major revision at a later time.